# Latent Variable Sequential Set Transformers for Joint Multi-Agent Motion Prediction

**Roger Girgis**[1,2] **Florian Golemo**[2,3] **Felipe Codevilla**[2,4] **Martin Weiss**[1,2]
**Jim Aldon D'Souza**[5] **Samira E. Kahou**[2,6,7,8] **Felix Heide**[5,9] **Christopher Pal**[1,2,3,8]

[1]Polytechnique Montréal, [2]Mila, Quebec AI Institute [3]ElementAI / Service Now,
[4]Independent Robotics, [5]Algolux, [6]McGill University, [7]École de technologie supérieure,
[8]Canada CIFAR AI Chair, [9]Princeton University. Correspondence: roger.girgis@gmail.com
https://fgolemo.github.io/autobots/

## Abstract

Robust multi-agent trajectory prediction is essential for the safe control of robotic systems. A major challenge is to efficiently learn a representation that approximates the true joint distribution of contextual, social, and temporal information to enable planning. We propose Latent Variable Sequential Set Transformers which are encoder-decoder architectures that generate scene-consistent multi-agent trajectories. We refer to these architectures as "AutoBots". The encoder is a stack of interleaved temporal and social multi-head self-attention (MHSA) modules which alternately perform equivariant processing across the temporal and social dimensions. The decoder employs learnable seed parameters in combination with temporal and social MHSA modules allowing it to perform inference over the entire future scene in a single forward pass efficiently. AutoBots can produce either the trajectory of one ego-agent or a distribution over the future trajectories for all agents in the scene. For the single-agent prediction case, our model achieves top results on the global nuScenes vehicle motion prediction leaderboard, and produces strong results on the Argoverse vehicle prediction challenge. In the multi-agent setting, we evaluate on the synthetic partition of TrajNet++ dataset to showcase the model's socially-consistent predictions. We also demonstrate our model on general sequences of sets and provide illustrative experiments modelling the sequential structure of the multiple strokes that make up symbols in the Omniglot data. A distinguishing feature of AutoBots is that all models are trainable on a single desktop GPU (1080 Ti) in under 48h.

## 1 Introduction

Many problems require processing complicated hierarchical compositions of sequences and sets. For example, multiple choice questions are commonly presented as a query (e.g. natural language) and an unordered set of options (Kembhavi et al., 2017; Richardson et al., 2013). Machine learning models that perform well in this setting should be insensitive to the order in which the options are presented. Similarly, for motion prediction tasks (Biktairov et al., 2020; Sadeghian et al., 2018) where the inputs are agent trajectories evolving over time and outputs are future agent trajectories, models should be insensitive to agent ordering.

In this work, we focus on the generative modelling of sequences and sets and demonstrate this method on a variety of different motion forecasting tasks. Suppose we have a set of $M \in \mathbb{N}$ sequences $\mathbb{X} = \{(x_1, \ldots, x_K)_1, \ldots, (x_1, \ldots, x_K)_M\}$ each with $K \in \mathbb{N}$ elements. Allowing also that $\mathbb{X}$ evolves across some time horizon $T$, we denote this sequence of sets as $\mathbf{X} = (\mathbb{X}_1, \ldots, \mathbb{X}_T)$.

This problem setting often requires the model to capture high-order interactions and diverse futures. Generally, solutions are built with auto-regressive sequence models which include $t$ steps of context. In our approach, to better model the multi-modal distribution over possible futures, we also allow to include a discrete latent variable $Z$, to create sequential models of sets of the form

$$P(\mathbb{X}_{t+1}|\mathbb{X}_t, \ldots, \mathbb{X}_1) = \sum_Z P(\mathbb{X}_{t+1}, Z|\mathbb{X}_t, \ldots, \mathbb{X}_1).$$

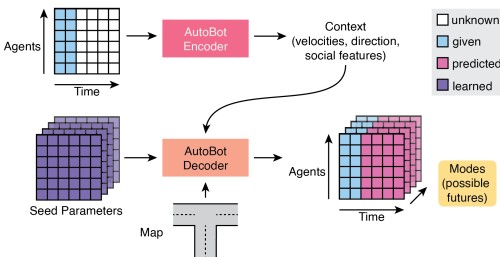

Figure 1: **Overview:** Our model generates multiple possible futures for agents based on several input timesteps. The input trajectories are encoded into a context tensor that captures their respective behavior and interaction with surrounding agents. While Transformer-based architectures are often autoregressive, we decode all future steps at once. This is accomplished via learnable seed parameters. Our decoder transforms these, together with the map and the context into the possible future trajectories that we call "modes".

In this work, we propose this type of parameterized conditional mixture model over sequences of sets, and call this model a Latent Variable Sequential Set Transformer (affectionately referred to as "AutoBot"). We evaluate AutoBot on *nuScenes* (Caesar et al., 2020) and *Argoverse* (Chang et al., 2019), two autonomous driving trajectory prediction benchmarks, on the synthetic partition of the *TrajNet++* (Kothari et al., 2021) dataset, a pedestrian trajectory forecasting benchmark, and *Omniglot* (Lake et al., 2015), a dataset of hand-drawn characters that we use for predicting strokes. All of these provide empirical validation of our model's ability to perform robust multi-modal prediction. Specifically, we make the following contributions:

- We propose a novel approach for modelling sequences of set-structured continuous variables, and extend this to a latent-variable formulation to capture multi-modal distributions.

- We provide a theoretical analysis highlighting our model's permutation sensitivity with respect to different components of the input data structures.

- We validate our method with strong empirical results on diverse tasks, including the large-scale *nuScenes* and *Argoverse* datasets for autonomous driving (Caesar et al., 2020; Chang et al., 2019), the multi-lingual *Omniglot* dataset of handwritten characters (Lake et al., 2015), and the synthetic partition of the *TrajNet++* pedestrian trajectory prediction task (Sadeghian et al., 2018).

## 2 BACKGROUND

We now review several components of the Transformer (Vaswani et al., 2017) and Set Transformer (Lee et al., 2019) architectures, mostly following the notation found in their manuscripts. For additional background, see their works and Appendix A.

**Multi-Head Self Attention (MHSA)** can be thought of as an information retrieval system, where a query is executed against a key-value database, returning values where the key matches the query best. While (Vaswani et al., 2017) defines MHSA on three tensors, for convenience, we input a single set-valued argument. Internally, MHSA then performs intra-set attention by casting the input set $\mathbb{X}$ to query, key and value matrices and adding a residual connection, $\text{MHSA}(\mathbb{X}) = \mathbf{X} + \text{MHSA}(\mathbf{X}, \mathbf{X}, \mathbf{X})$.

**Multi-Head Attention Blocks (MAB)** resemble the encoder proposed in Vaswani et al. (2017), but lack the position encoding and dropout (Lee et al., 2019). Specifically, they consist of the MHSA operation described in Eq. 8 (appendix) followed by a row-wise feed-forward neural network (rFFN), with residual connections and layer normalization (LN) (Ba et al., 2016) after each block. Given an input set $\mathbb{X}$ containing $n_x$ elements of dimension $d$, and some conditioning variables $\mathbb{C}$ containing $n_c$ elements of dimension $d$, the MAB can be described by the forward computation,

$$\text{MAB}(\mathbf{X}) = \text{LN}(\mathbf{H} + \text{rFFN}(\mathbf{H})), \quad \text{where} \quad \mathbf{H} = \text{LN}(\mathbf{X} + \text{MHSA}(\mathbf{X}, \mathbf{X}, \mathbf{X})). \tag{1}$$

**Multi-head Attention Block Decoders (MABD)** were also introduced in Vaswani et al. (2017), and were used to produce decoded sequences. Given input matrices $\mathbf{X}$ and $\mathbf{Y}$ representing sequences, the decoder block performs the following computations:

$$\text{MABD}(\mathbf{Y}, \mathbf{X}) = \text{LN}(\mathbf{H} + \text{rFFN}(\mathbf{H}))$$
$$\text{where} \quad \mathbf{H} = \text{LN}(\mathbf{H}' + \text{MHSA}(\mathbf{H}', \mathbf{X}, \mathbf{X})) \tag{2}$$
$$\text{and} \quad \mathbf{H}' = \text{LN}(\text{MHSA}(\mathbf{Y}))$$

Our model, described throughout the following section, makes extensive use of these functions.

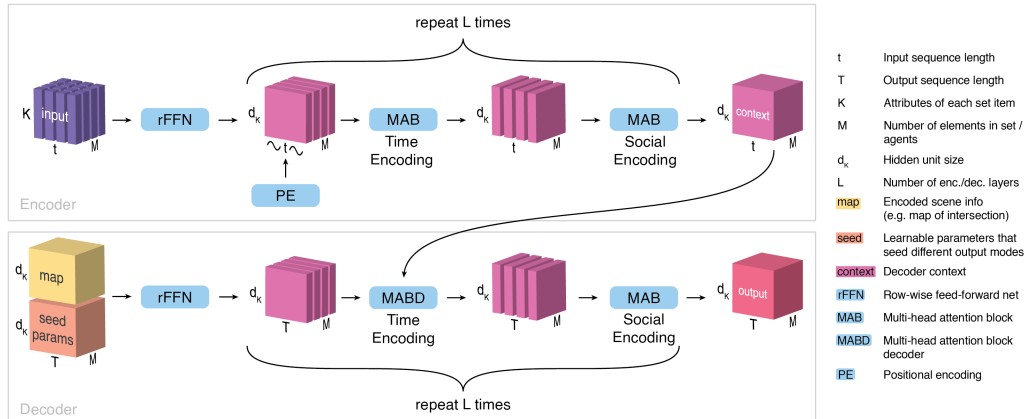

Figure 2: **Architecture Overview.** Our model takes as input a tensor of dimension $K, M, t$. A row-wise feed-forward network (rFFN) is applied to each row along the $t \times M$ plane transforming vectors of dimension $K$ to $d_K$. After adding positional encoding (PE) to the $t$ axis, the encoder passes the tensor through $L$ repeated layers of multi-head attention blocks (MAB) that are applied to the time axis (time encoding) and the agent axis (social encoding) before outputting the context tensor. In the decoder, the encoded map and the learnable seed parameters tensor are concatenated and passed through an rFFN before being passed through $L$ repeated layers of a multi-head attention block decoder (MABD) along the time axis (using the context from the encoder) followed by a MAB along the agent axis. This figure shows the process for predicting one possible future trajectory ("mode"). When multiple modes are predicted, each mode has its own learnable seed parameters and the decoder is rolled out once for each mode.

## 3 LATENT VARIABLE SEQUENTIAL SET TRANSFORMERS

Latent Variable Sequential Set Transformers is a class of encoder-decoder architectures that process sequences of sets (see Fig 2). In this section, we describe the encoding and decoding procedures, the latent variable mechanism, and the training objective. Additionally, we provide proof of the permutation equivariance of our model in Appendix B, and implementation details in Appendix C as well as a discussion of the special case of AutoBot-Ego, which is similar to AutoBots but predicts a future sequence for only one element in a scene.

### 3.1 ENCODER: INPUT SEQUENCE REPRESENTATION

AutoBot takes as an input a sequence of sets, $\mathbf{X}_{1:t} = (\mathbb{X}_1, \ldots, \mathbb{X}_t)$, which in the motion prediction setting may be viewed as the state of a scene evolving over $t$ timesteps. Let us describe each set as having $M$ elements (or agents) with $K$ attributes (e.g. x/y position). To process the social and temporal information the encoder applies the following two transformations. Appendix B.2 provides a proof of equivariance with respect to the agent dimension for this encoder.

First, the AutoBots encoder injects temporal information into the sequence of sets using a sinusoidal positional encoding function PE(.) (Vaswani et al., 2017). For this step, we view the data as a collection of matrices, $\{\mathbf{X}_0, \ldots, \mathbf{X}_M\}$, that describe the temporal evolution of agents. The encoder processes temporal relationships between sets using a MAB, performing the following operation on each set where $m \in \{1, \ldots, M\}$:

$$\mathbf{S}_m = \text{MAB}(\text{rFFN}(\mathbf{X}_m))$$

where rFFN(.) is an embedding layer projecting each element in the set to the size of the hidden state $d_K$. Next, we process temporal slices of $\mathbf{S}$, retrieving sets of agent states $\mathbb{S}_\tau$ at some timestep $\tau$ and is processed using $\mathbb{S}_\tau = \text{MAB}(\mathbb{S}_\tau)$. These two operations are repeated $L_{\text{enc}}$ times to produce a **context tensor** $\mathbf{C} \in \mathbb{R}^{(d_K, M, t)}$ summarizing the entire input scene, where $t$ is the number of timesteps in the input scene.

### 3.2 DECODER: MULTIMODAL SEQUENCE GENERATION

The goal of the decoder is to generate temporally and socially consistent predictions in the context of multi-modal data distributions. To generate $c \in \mathbb{N}$ different predictions for the same input scene,

the AutoBot decoder employs $c$ matrices of learnable seed parameters $\mathbf{Q}_i \in \mathbb{R}^{(d_K, T)}$ where $T$ is the prediction horizon, where $i \in \{1, ..., c\}$. Intuitively, each matrix of learnable seed parameters corresponds to a setting of the discrete latent variable in AutoBot. Each learnable matrix $\mathbf{Q}_i$ is then repeated across the agent dimension $M$ times to produce the input tensor $\mathbf{Q}'_i \in \mathbb{R}^{(d_K, M, T)}$. Additional contextual information (such as a rasterized image of the environment) is encoded using a convolutional neural network to produce a vector of features $\mathbf{m}_i$ where $i \in \{1, ..., c\}$. In order to provide the contextual information to all future timesteps and to all set elements, we copy this vector along the $M$ and $T$ dimensions, producing the tensor $\mathbf{M}_i \in \mathbb{R}^{(d_K, M, T)}$. Each tensor $\mathbf{Q}'_i$ is concatenated with $\mathbf{M}_i$ along the $d_K$ dimension, as shown on the left side of Figure 2 (bottom). This tensor is then processed using a rFFN(.) to produce the tensor $\mathbf{H} \in \mathbb{R}^{(d_K, M, T)}$.

We begin to decode by processing the time dimension, conditioning on the encoder's output $\mathbf{C}$, and the encoded seed parameters and environmental information in $\mathbf{H}$. The decoder processes each of $m$ agents in $\mathbf{H}$ separately with an MABD layer:

$$\mathbf{H}'_m = \text{MABD}(\mathbf{H}_m, \mathbf{C}_m)$$

where $\mathbf{H}'$ is the output tensor that encodes the future time evolution of each element in the set independently. In order to ensure that the future scene is socially consistent between set elements, we process each temporal slice of $\mathbf{H}'$, retrieving sets of agent states $\mathbb{H}'_\tau$ at some future timestep $\tau$ where each element $\boldsymbol{h}'_\tau \in \mathbb{H}_\tau$ is processed by a MAB layer. Essentially, the MAB layer here performs a per-timestep attention between all set elements, similar to the Set Transformer (Lee et al., 2019).

These two operations are repeated $L_{\text{dec}}$ times to produce a final output tensor for mode $i$. The decoding process is repeated $c$ times with different learnable seed parameters $\mathbf{Q}_i$ and additional contextual information $\mathbf{m}_i$. The output of the decoder is a tensor $\mathbf{O} \in \mathbb{R}^{(d_K, M, T, c)}$ which can then be processed element-wise using a neural network $\phi(.)$ to produce the desired output representation. In several of our experiments, we generate trajectories in $(x, y)$ space, and as such, $\phi$ produces the parameters of a bivariate Gaussian distribution.

**Note on the learnable seed parameters.** One of the main contributions that make AutoBot's inference and training time faster than prior work is the use of the decoder seed parameters $\mathbf{Q}$. These parameters serve a dual purpose. Firstly, they account for the diversity in future prediction, where each matrix $\mathbf{Q}_i \in \mathbb{R}^{(d_k, T)}$ (where $i \in \{1, ..., c\}$) corresponds to one setting of the discrete latent variable. Secondly, they contribute to the speed of AutoBot by allowing it to perform inference on the entire scene with a single forward pass through the decoder without sequential sampling. Prior work (e.g., Tang and Salakhutdinov (2019)) uses auto-regressive sampling of the future scene with interlaced social attention, which allows for socially consistent multi-agent predictions. Other work (e.g., Messaoud et al. (2020)) employ a single output MLP to generate the entire future of each agent *independently* with a single forward pass. Using the attention mechanisms of transformers, AutoBot combines the advantages of both approaches by deterministically decoding the future scene starting from each seed parameter matrix. In Appendix C.5, we compare the inference speed of AutoBots with their autoregressive counterparts.

### 3.3 TRAINING OBJECTIVE

Given a dataset $\mathcal{D} = \left\{(\mathbb{X}_1, \ldots, \mathbb{X}_T)\right\}_{i=1}^{N}$ consisting of $N$ sequences, each having $T$ sets, our goal is to maximize the likelihood of the future trajectory of all elements in the set given the input sequence of sets, i.e., $\max p(\mathbf{X}_{t+1:T}|\mathbf{X}_{1:t})$. In order to simplify notation, we henceforth refer to the ego-agent's future trajectory by $\mathcal{Y} = \mathbf{X}_{t+1:T}$. As discussed above, AutoBot employs discrete latent variables, which allows us to compute the log-likelihood exactly. The gradient of our objective function can then be expressed as follows:

$$\nabla_\theta \mathcal{L}(\theta) = \nabla_\theta \log p_\theta(\mathcal{Y}|\mathbf{X}_{1:t}) = \nabla_\theta \log \left( \sum_Z p_\theta(\mathcal{Y}, Z|\mathbf{X}_{1:t}) \right)$$
$$= \sum_Z p_\theta(Z|\mathcal{Y}, \mathbf{X}_{1:t}) \nabla_\theta \log p_\theta(\mathcal{Y}, Z|\mathbf{X}_{1:t}) \tag{3}$$

As previously discussed in Tang and Salakhutdinov (2019), computing the posterior likelihood $p_\theta(Z|\mathcal{Y}, \mathbf{X}_{1:t})$ is difficult in general and varies with $\theta$. As discussed in Bishop (2006), one can introduce a distribution over the latent variables, $q(Z)$, that is convenient to compute. With this

distribution, we can now decompose the log-likelihood as follows:

$$\log p_\theta(\mathcal{Y}|\mathbf{X}_{1:t}) = \sum_Z q(Z) \log \frac{p_\theta(\mathcal{Y}, Z|\mathbf{X}_{1:t})}{q(Z)} + D_{KL}(q(Z)||p_\theta(Z|\mathcal{Y}, \mathbf{X}_{1:t})), \quad (4)$$

where $D_{KL}(.||.)$ is the Kullback-Leibler divergence between the approximating posterior and the actual posterior. A natural choice for this approximating distribution is $q(Z) = p_{\theta_{old}}(Z|\mathcal{Y}, \mathbf{X}_{1:t})$, where $\theta_{old}$ corresponds to AutoBot's parameters before performing the parameter update. With this choice of $q(Z)$, the objective function can be re-written as

$$\mathcal{L}(\theta) = Q(\theta, \theta_{old}) + \text{const.}$$
$$Q(\theta, \theta_{old}) = \sum_Z p_{\theta_{old}}(Z|\mathcal{Y}, \mathbf{X}_{1:t}) \log p_\theta(\mathcal{Y}, Z|\mathbf{X}_{1:t})$$
$$= \sum_Z p_{\theta_{old}}(Z|\mathcal{Y}, \mathbf{X}_{1:t}) \big\{ \log p_\theta(\mathcal{Y}|Z, \mathbf{X}_{1:t}) + \log p_\theta(Z|\mathbf{X}_{1:t}) \big\}. \quad (5)$$

where the computation of $p_\theta(Z|\mathbf{X}_{1:t})$ is described in Appendix C.2. Note that the posterior $p_{\theta_{old}}(Z|\mathcal{Y}, \mathbf{X}_{1:t})$ can be computed exactly in the model we explore here since AutoBot operates with discrete latent variables. Therefore, our final objective function becomes to maximize $Q(\theta, \theta_{old})$ and minimize $D_{KL}(p_{\theta_{old}}(Z|\mathcal{Y}, \mathbf{X}_{1:t})||p_\theta(Z|\mathbf{X}_{1:t}))$.

As mentioned, for our experiments we use a function $\phi$ to produce a bivariate Gaussian distribution for each agent at every timestep. We found that to ensure that each mode's output sequence does not have high variance, we introduce a mode entropy (ME) regularization term to penalize large entropy values of the output distributions,

$$L_{\text{ME}} = \lambda_e \max_Z \sum_{\tau=t+1}^{T} H(p_\theta(\mathbb{X}_\tau|Z, \mathbf{X}_{1:t})). \quad (6)$$

As we can see, this entropy regularizer penalizes only the mode with the maximum entropy. Our results in Section D.2 show the importance of this term in learning to cover the modes of the data.

## 4 EXPERIMENTS

We now demonstrate the capability of AutoBots to model sequences of set-structured data across several domains. In subsection 4.1 we show strong performance on the competitive motion-prediction dataset nuScenes (Caesar et al., 2020), a dataset recorded from a self-driving car, capturing other vehicles and the intersection geometry. In subsection 4.2, we demonstrate our model on the Argoverse autonomous driving challenge. In subsection 4.3, we show results on the synthetic partition of the TrajNet++ (Sadeghian et al., 2018) dataset which contains multi-agent scenes with a high degree of interaction between agents. Finally in subsection 4.4, we show that our model can be applied to generate characters conditioning across strokes, and verify that AutoBot-Ego is permutation-invariant across an input set. See Appendix D.2, for an illustrative toy experiment that highlights our model's ability to capture discrete latent modes.

### 4.1 NUSCENES, A REAL-WORLD DRIVING DATASET

In this section, we present AutoBot-Ego's results on the nuScenes dataset. The goal of this benchmark dataset is to predict the ego-agent's future trajectory (6 seconds at 2hz) given the past trajectory (up to 2 seconds at 2hz). Additionally, the benchmark provides access to all neighbouring agents' past trajectories and a birds-eye-view RGB image of the road network in the vicinity of the ego-agent. Figure 3 shows three example predictions produced by AutoBot-Ego trained with $c = 10$. We observe that the model learns to produce trajectories that are in agreement with the road network and covers most possible futures directions. We also note that for each direction, AutoBot-Ego assigns different modes to different speeds to more efficiently cover the possibilities.

We compare our results on this benchmark to the state-of-the-art in Table 1. **Min ADE (5)** and **(10)** are the average of pointwise L2 distances between the predicted trajectory and ground truth over 5 and 10 most likely predictions respectively. A prediction is classified as a *miss* if the maximum pointwise L2 distance between the prediction and ground truth is greater than 2 meters. **Miss Rate Top-5 (2m)** and **Top-10 (2m)** are the proportion of misses over all agents, where for each agent,

| Model | Min ADE (5) | Min ADE (10) | Miss Rate Top-5 (2m) | Miss Rate Top-10 (2m) | Min FDE (1) | Off Road Rate |
|---|---|---|---|---|---|---|
| Noah_prediction | 1.59 | 1.37 | 0.69 | 0.62 | 9.23 | 0.08 |
| CXX | 1.63 | 1.29 | 0.69 | 0.60 | 8.86 | 0.08 |
| LISA(MHA_JAM) | 1.81 | 1.24 | 0.59 | 0.46 | 8.57 | 0.07 |
| Trajectron++ | 1.88 | 1.51 | 0.70 | 0.57 | 9.52 | 0.25 |
| CoverNet | 2.62 | 1.92 | 0.76 | 0.64 | 11.36 | 0.13 |
| Physics Oracle | 3.70 | 3.70 | 0.88 | 0.88 | 9.09 | 0.12 |
| WIMP | 1.84 | 1.11 | **0.55** | **0.43** | 8.49 | 0.04 |
| GOHOME | 1.42 | 1.15 | 0.57 | 0.47 | **6.99** | 0.04 |
| AutoBot-Ego (c=10) | 1.43 | 1.05 | 0.66 | 0.45 | 8.66 | 0.03 |
| AutoBot-Ego (ensemble) | **1.37** | **1.03** | 0.62 | 0.44 | 8.19 | **0.02** |

Table 1: **Quantitative Results on the nuScenes dataset**. Other methods: LISA (Messaoud et al., 2020); Trajectron++ (Salzmann et al., 2020); CoverNet (Phan-Minh et al., 2020); Physics Oracle (Caesar et al., 2020); WIMP (Khandelwal et al., 2020); GOHOME (Gilles et al., 2021)

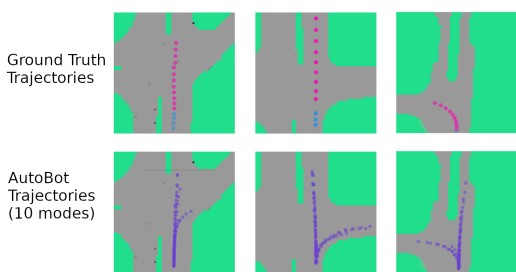

Ground Truth Trajectories

AutoBot Trajectories (10 modes)

Figure 3: **NuScenes Trajectory Prediction**. Top row: birds-eye-view of road network with ground truth trajectory data where given trajectory is cyan and held-out trajectory information is pink. Bottom row: diverse trajectories generated by the different modes of AutoBot-Ego. The model generates trajectories that adhere to the road network and captures distinct possible futures.

5 and 10 most likely predictions respectively are evaluated to check if they're misses. **Min FDE (1)** is the L2 distance between the final points of the prediction and ground truth of the most likely prediction averaged over all agents. **Off Road Rate** is the fraction of predicted trajectories that are not entirely contained in the drivable area of the map.

We can see that AutoBot-Ego obtains the best performance on the Min ADE (10) metric and on Off Road Rate metric, and strong performance on the other metrics. Furthermore, using an ensemble of three AutoBot-Ego models slightly improves the performance. This demonstrates our model's ability to capture multi-modal distributions over future trajectories due to the latent variables. AutoBot-Ego excelled in maintaining predictions on the road, showing that the model has effectively used its capacity to attend to the map information. Finally, we note that like all other models, AutoBot-Ego struggles to predict the probability of the correct mode, which results in poor performance of the Min FDE (1). The most interesting part of AutoBot-Ego is its computational requirements. We highlight that AutoBot-Ego was trained on a single Nvidia GTX 1080 Ti for 3 hours. This is in large part due to the use of the decoder seed parameters which allow us to predict the future trajectory of the ego-agent in a single forward pass. Note that in Appendix C.5, we present the inference speed of AutoBot-Ego compared to an auto-regressive version. Ablation studies on various components of AutoBot-Ego are also presented in Appendix D.1.

## 4.2 ARGOVERSE RESULTS

Table 2 shows the results of AutoBot-Ego (with $c = 6$) on the Argoverse test set. While our method does not achieve top scores, it uses much less computation, less data augmentation and no ensembling compared to alternatives. Further, we compare quite well to the 2020 Argoverse competition winner, a model called "Jean". In addition, we note that the Argoverse challenge does not restrict the use of additional training data and that many of the top entries on the leaderboard have no publicly available papers or codebases. Many methods are therefore optimized for Argoverse by means of various data augmentations. For example, Scene Transformer (Ngiam et al., 2021) used a data augmentation scheme that includes agent dropout and scene rotation which we have not, and the current top method and winner of the 2021 challenge (QCraft) makes use of ensembling.

| Model | Min ADE (↓) | Min FDE (↓) | Miss Rate (↓) | DAC (↑) |
|---|---|---|---|---|
| AutoBot-Ego (Valid Set) | 0.73 | 1.10 | 0.12 | - |
| AutoBot-Ego (Test Set) | 0.89 (top-5) | 1.41 (top-5) | 0.16 (top-6) | 0.9886 (top-3) |
| Jean (2020 winner) | 0.97 | 1.42 | 0.13 | 0.9868 |
| WIMP | 0.90 | 1.42 | 0.17 | 0.9815 |
| TNT | 0.94 | 1.54 | 0.13 | 0.9889 |
| LaneGCN | 0.87 | 1.36 | 0.16 | 0.9800 |
| TPCN | 0.85 | 1.35 | 0.16 | 0.9884 |
| mmTransformer | 0.84 | 1.34 | 0.15 | 0.9842 |
| GOHOME | 0.94 | 1.45 | **0.105** | 0.9811 |
| Scene Transformer | **0.80** | **1.23** | 0.13 | **0.9899** |

Table 2: **Quantitative Results on the Argoverse test set**. Other methods that had an associated paper: Jean (Mercat et al., 2020); WIMP (Khandelwal et al., 2020); TNT (Zhao et al., 2020a); LaneGCN (Liang et al., 2020); TPCN (Ye et al., 2021); mmTransformer (Liu et al., 2021); GOHOME (Gilles et al., 2021); Scene Transformer (Ngiam et al., 2021).

| Model | Ego Agent's Min ADE(6) (↓) | Number Of Collisions (↓) | Scene-level Min ADE(6) (↓) | Scene-level Min FDE(6) (↓) |
|---|---|---|---|---|
| Linear Extrapolation | 0.439 | 2220 | 0.409 | 0.897 |
| AutoBot-AntiSocial | 0.196 | 1827 | 0.316 | 0.632 |
| AutoBot-Ego | 0.098 | 1144 | 0.214 | 0.431 |
| AutoBot | **0.095** | **139** | **0.128** | **0.234** |

Table 3: **TrajNet++ ablation studies for a multi-agent forecasting scenario.** We investigate the impact of using the social attention in the encoder and in the decoder. We found that AutoBot is able to cause significantly fewer collisions between agents compared to variants without the social attention, and improves the prediction accuracy on the scene-level metrics while maintaining strong performance on the ego-agent. Metrics are reported on the validation set.

We believe that a fair comparison of the performance between neural network methods should ensure that each architecture is parameter-matched and the training is FLOP-matched. While the Argoverse leaderboard does not require entrants to report their parameter or FLOP count, we report available public statistics to enable comparison here. First, Scene Transformer (Ngiam et al., 2021) reports that their model trains in 3 days on TPUs and TPU-v3s have a peak performance of 420 TeraFLOPS, which means their model could have consumed up to 3 days × 24 hours × 420 TFLOPS, or 108,000,000 TFLOPS. On the other hand, AutoBot-Ego was trained on a Nvidia 1080ti for 10 hours, consuming roughly 396,000 TFLOPs, or 0.4% as much. Jean's method (Mercat et al., 2020) consumes roughly 8,600,000 TFLOPs (training our method uses only 5% of that compute). While the method currently topping the leaderboard has not revealed their model details, the fact that they used ensembles very likely requires significantly more compute than our method.

### 4.3 TRAJNET++ SYNTHETIC DATASET

In this section, we demonstrate the utility of the having social and temporal attention in the encoder and decoder of AutoBot. We chose to evaluate on the synthetic partition of the TrajNet++ dataset since this was specifically designed to have a high-level of interaction between scene agents (Kothari et al., 2021). In this task, we are provided with the state of all agents for the past 9 timesteps and are tasked with predicting the next 12 timesteps for all agents. The scene is initially normalized around one (ego) agent such that the agent's final position in the input sequence is at $(0, 0)$ and its heading is aligned with the positive $y$ direction. There are a total of $54,513$ unique scenes in the dataset, which we split into $49,062$ training scenes and $5,451$ validation scenes. For this experiment, we apply our general architecture (AutoBot) to multi-agent scene forecasting, and perform an ablation study on the social attention components in the encoder and the decoder.

Table 3 presents three variants of the model, and a baseline extrapolation model which shows that the prediction problem cannot be trivially solved. The first, AutoBot-AntiSocial, corresponds to a model without the intra-set attention functions in the encoder and decoder (see Figure 2). The second variant, AutoBot-Ego, only sees the past social context and has no social attention during

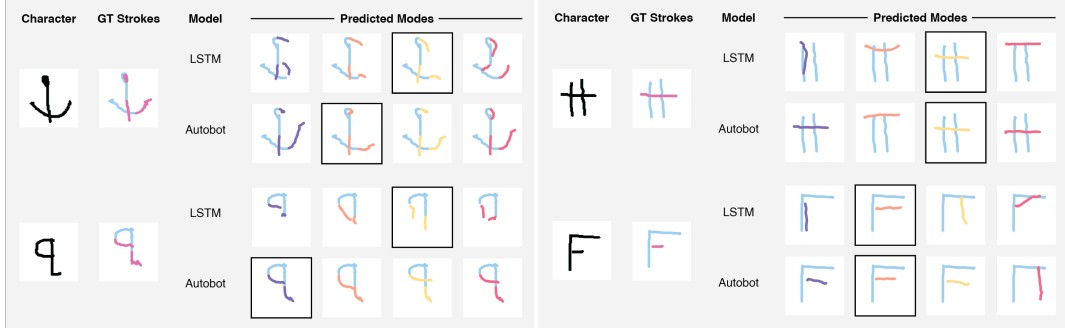

Figure 4: **Omniglot qualitative results.** *Left: stroke completion task.* We show examples of characters generated using AutoBot and an LSTM baseline. The first two columns show the ground-truth image of the character and the corresponding ground-truth strokes. In this task, the models are provided with the first half of all strokes (cyan) and are tasked with generating the other half (pink). The four other columns show the generated future strokes, one for each latent mode. We can see that AutoBot produces more consistent and realistic characters compared to the LSTM baseline. *Right: character completion task.* We show two examples of characters completed using AutoBot-Ego and an LSTM baseline. The first two columns show the ground-truth image of the character as before. In this case, the models are provided with two complete strokes and are tasked with generating a new stroke to complete the character. We observe that AutoBot-Ego generates more realistic characters across all modes, given ambiguous input strokes.

the trajectory generation process. The third model corresponds to the general architecture, AutoBot. All models were trained with $c = 6$ and we report scene-level error metrics as defined by Casas et al. (2020). Using the social attention in the decoder significantly reduces the number of collisions between predicted agents' futures and significantly outperforms its ablated counterparts on scene-level minADE and minFDE. We expect this at a theoretical level since the ego-centric formulation makes the independence assumption that the scene's future rollout can be decomposed as the product of the individual agent's future motion in isolation, as remarked in Casas et al. (2020). AutoBot does not make this assumption as it forecasts the future of all agents jointly. We refer the reader to Appendix D.3 for additional ablation experiments and qualitative results on this dataset.

## 4.4 OMNIGLOT, STROKE-BASED IMAGE GENERATION

To demonstrate the general efficacy of our approach to sequence generation, we demonstrate AutoBots on a character completion task, a diverse alternative to automotive and pedestrian trajectory prediction. We set up two different tasks on the Omniglot dataset (Lake et al., 2015): (a) **stroke-completion task** in which the model is presented with the first half of each stroke and has to complete all of them in parallel and (b) **character-completion task** in which the model is given several full strokes and has to draw the last stroke of the character. In task (a), the model is trained on the entire Omniglot dataset (training set), in which strokes are encoded as sequential sets of points, and we are comparing the performance of AutoBot qualitatively against that of an LSTM baseline with social attention and discrete latents (similar to AutoBot but with classic RNN architecture) on a held-out test set. With the first task, we illustrate that our model is flexible enough to learn to legibly complete characters from a variety of different languages. In task (b), we train AutoBot-Ego only on characters "F", "H", and "Π" to demonstrate that our discrete latent variable captures plausible character completions given the context of the other strokes, e.g. a character "F" can be completed as an "H" when the top stroke is missing. The results of these experiments can be found in Fig. 4. Implementation details and additional results can be found in Appendix section D.5.

## 5 RELATED WORK

**Processing set structured data.** Models of set-valued data (i.e. unordered collections) should be permutation-invariant (see Appendix B for formal definitions), and be capable of processing sets with arbitrary cardinality. Canonical forms of feed forward neural networks and recurrent neural network models do not have these properties (Lee et al., 2019). One type of proposed solution is to integrate pooling mechanisms with feed forward networks to process set data (Zaheer et al., 2017; Su et al., 2015; Hartford et al., 2016). This is referred as social pooling in the context of motion prediction (Alahi et al., 2016; Deo and Trivedi, 2018; Lee et al., 2017). Attention-based models (Lee et al.,

2019; Casas et al., 2020; Yuan et al., 2021; Ngiam et al., 2021; Zhao et al., 2020b) or graph-based models (Zhao et al., 2020a; Messaoud et al., 2020; Li et al., 2020b; Gao et al., 2020) have performed well in set processing tasks that require modelling high-order element interactions. In contrast, we provide a joint single pass attention-based approach for modelling more complex data-structures composed by sets **and** sequence, while also addressing how to generate diverse and heterogeneous data-structures. Although those properties exist in concurrent motion prediction papers (Tang and Salakhutdinov, 2019; Casas et al., 2020; Yuan et al., 2021; Ngiam et al., 2021; Li et al., 2020a), our proposed method is targeting a more general set of problems while still getting competitive results in motion prediction benchmarks at a fraction of other models' compute.

**Diverse set generation.** Generation of diverse samples using neural sequence models is an important part of our model and a well-studied problem. For example, in Bowman et al. (2015) a variational (CVAE) approach is used to construct models that condition on attributes for dialog generation. The CVAE approach has also been integrated with transformer networks (Lin et al., 2020; Wang and Wan, 2019) for diverse response generation. For motion prediction, Lee et al. (2017) and Deo and Trivedi (2018) use a variational mechanism where each agent has their own intentions encoded. Li et al. (2020a) output Gaussian mixtures at every timestep and sample from it to generate multiple futures. The most similar strategy to ours was developed by Tang and Salakhutdinov (2019). Other recent works (Suo et al., 2021; Casas et al., 2020) use continuous latent variables to model the multiple actors' intentions jointly. We built on this method by using one joint transformer conditioned on the latent variables and added a maximum entropy likelihood loss that improves the variability of the generated outcomes.

**Sequential state modelling.** We are interested in data that can be better represented as a sequence. The main example of this is language modelling (Vaswani et al., 2017; Devlin et al., 2018), but also includes sound (Kong et al., 2020), video (Becker et al., 2018), and others (Kant et al., 2020; Yang et al., 2021). For motion prediction, prior work has largely focused on employing RNNs to model the input and/or output sequence (Alahi et al., 2016; Lee et al., 2017; Deo and Trivedi, 2018; Tang and Salakhutdinov, 2019; Messaoud et al., 2020; Casas et al., 2020; Li et al., 2020b;a). With its recent success in natural language processing, Transformers (Vaswani et al., 2017) have been adopted by recent approaches for motion forecasting. Yu et al. (2020) uses a graph encoding strategy for pedestrian motion prediction. Giuliari et al. (2021) and Liu et al. (2021) propose a transformer-based encoding to predict the future trajectories of agents separately. In contrast, AutoBots produces representations that jointly encode both the time and social dimensions by treating this structure as a sequence of sets. This provides reuse of the model capacity and a lightweight approach. Concurrent to our method, Yuan et al. (2021), also jointly encodes social and time dimensions but flattens both of those axis into a single dimension.

## 6 CONCLUSION

In this paper, we propose the Latent Variable Sequential Set Transformer to model time-evolution of sequential sets using discrete latent variables. We make use of the multi-head attention block to efficiently perform intra-set attention, to model the time-dependence between the input and output sequence, and to predict the prior probability distribution over the latent modes. We validate our AutoBot-Ego by achieving competitive results on the nuScenes benchmark for ego-centric motion forecasting in the domain of autonomous vehicles. We further demonstrate that AutoBot can model diverse sequences of sets that adhere to social conventions. We validate that since our model can attend to the hidden state of all agents during the generative process, it produces scene-consistent predictions on the TrajNet++ dataset when compared with ego-centric forecasting methods.

**Limitations:** Our approach is limited to operate in a setting where a perceptual pipeline is providing structured sets representing the other entities under consideration. If our approach is used in a robot or autonomous car, any errors or unknown entities not captured by sets used as input to the algorithm would not be modelled by our approach. Memory limitations may also limit the number of sets that can be processed at any time. In future work, we plan to extend this line of work to broader and more interpretable scene-consistency. Further, we would like to investigate the combination of a powerful sequence model like AutoBot with a reinforcement learning algorithm to create a data-efficient model-based RL agent for multi-agent settings.

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

## A  BACKGROUND

We now review several components of the Transformer (Vaswani et al., 2017) and Set Transformer (Lee et al., 2019) architectures, mostly following the notation found in their manuscripts.

**Multi-Head Self Attention (MHSA)** as proposed in (Vaswani et al., 2017) is a function defined on $n_q$ query vectors $\mathbf{Q} \in \mathbb{R}^{n_q \times d_q}$, and key-value pairs ($\mathbf{K} \in \mathbb{R}^{n_k \times d_q}$ and $\mathbf{V} \in \mathbb{R}^{n_k \times d_v}$ ). With a single attention head, the function resolves the queries $\mathbf{Q}$ by performing the computation:

$$\text{Attn}(\mathbf{Q}, \mathbf{K}, \mathbf{V}) = \text{softmax}(\mathbf{Q}\mathbf{K}^{\top})\mathbf{V}, \tag{7}$$

where $d_q$ is the dimension of the query and key vectors and $d_v$ is the dimension of the value vector. MHSA consists of $h$ attention heads performing the operation shown in Eq. 7 with $h$ linear projections of keys, queries, and values. The final output is a linear projection of the concatenated output of each attention head. These computations can be expressed as

$$\text{MHSA}(\mathbf{Q}, \mathbf{K}, \mathbf{V}) = \text{concat}(\text{head}_1, \dots, \text{head}_h)\mathbf{W}^O, \quad \text{where}$$
$$\text{head}_i = \text{Attn}(\mathbf{Q}\mathbf{W}_i^Q, \mathbf{K}\mathbf{W}_i^K, \mathbf{V}\mathbf{W}_i^V), \tag{8}$$

and where $\mathbf{W}_i^Q$, $\mathbf{W}_i^V$ and $\mathbf{W}_i^K$ along with the output projection matrix $\mathbf{W}^O$ are the learnable projection matrices of each attention head in the MHSA. Note that $\mathbf{W}_i^Q$, $\mathbf{W}_i^V$ and $\mathbf{W}_i^K$ project the initial set of inputs to smaller dimensionality. For example, $\mathbf{W}_i^Q \in \mathbb{R}^{d_q \times d_q^M}$ projects the original queries with dimension $d_q$ to $d_q^M$ which is typically chosen to be $d_q/h$. This operation has useful applications in learning a representation of an input set where all its elements interact with each other.

An interesting application of MHSA is to perform self-attention on an input set (Lee et al., 2019). Given an input set $X$, one can perform intra-set attention by using $X$ as the queries, keys and values and having a residual connection, i.e., $\text{MHSA}(\mathbf{X}) = X + \text{MHSA}(\mathbf{X}, \mathbf{X}, \mathbf{X})$. This operation has useful applications in learning a representation of an input set where all its elements interact with each other.

**Multi-head Attention Block Decoders (MABD)** were also introduced in Vaswani et al. (2017), and were used to produce decoded sequences. Given an input matrix $\mathbf{X}$ and an output matrix $\mathbf{Y}$ representing sequences, the decoder block performs the following computations:

$$\text{MABD}(\mathbf{Y}, \mathbf{X}) = \text{LN}(\mathbf{H} + \text{rFFN}(\mathbf{H}))$$
$$\text{where} \quad \mathbf{H} = \text{LN}(\mathbf{H}' + \text{MHSA}(\mathbf{H}', \mathbf{X}, \mathbf{X})) \tag{9}$$
$$\text{and} \quad \mathbf{H}' = \text{LN}(\text{MHSA}(\mathbf{Y}))$$

During training, MABD can be trained efficiently by shifting $Y$ backward by one (with a start token at the beginning) and computing the output with one forward pass. During testing where one does not have access to $Y$, the model would then generate the future autoregressively. In order to avoid conditioning on the future during training, the $\text{MHSA}(Y)$ operation in MABD employs time-masking. This would prevent the model from accessing privileged future information during training.

## B  PERMUTATION EQUIVARIANCE OF THE MODEL

In this section, we prove the permutation equivariance of the Latent Variable Sequential Set Transformer with respect to the dimension $M$. To do so, we analyze the model's response to permutation along this dimension of the input tensor. This tensor, $\mathbf{X}$, is of dimension $(K, M, t)$, where $K, M, t \in \mathbb{N}$. The dimensions $t$ and $K$ are ordered, i.e. if we index into the tensor along dimension $M$ (we denote this operation as $\mathbf{X}_m$ where $m \in \{1, \dots, M\}$) then we retrieve a $(K, t)$-dimensional matrix. However, the $M$ dimension is unordered. This implies that when we index by another dimension, for example $t$, we retrieve an invertible multiset of vectors, denoted $\mathbb{X}_\tau = \{\boldsymbol{x}_1, \dots, \boldsymbol{x}_M\}$ where $\tau \in \{1, \dots, t\}$ and $\boldsymbol{x}_m \in \mathbb{X}_\tau$ are $K$-dimensional.

Our proof that our model is permutation equivariant on $M$ demonstrates a clear difference with the properties of similar models. For example, the model proposed in Set Transformers (Lee et al., 2019) features a permutation equivariant encoder combined with a permutation *invariant* decoder. And while Deep Sets (Zaheer et al., 2017) provides a theoretical framework for designing neural network

models that operating on sets, they do not directly address the processing of hetereogeneous data structures as we do here.

The rest of the section will proceed in the following manner:

1. Provide definitions and lemmas of the mathematical structures we will use in our proofs.
2. Show the permutation equivariance of the AutoBot encoder on $M$.
3. Show the permutation equivariance of the AutoBot decoder on $M$.

For clarity, we re-state the motivation for proving permutation equivariance in this particular model. In tasks like motion prediction, we may not be able to uniquely identify moving objects (perhaps due to occlusion or similar visual features). Therefore, we may wish to represent these objects as a collection of states evolving over time. Readers may benefit from this perspective on $\mathbf{X}$ as a time-evolving ($t$) collection of $M$ objects with $K$ attributes. We described $\mathbb{X}_\tau$ as a multiset for generality, as it permits repeated elements. However, in motion prediction tasks it might be safe to assume that no two distinct objects will share the same attribute vector. By proving that our model is equivariant with respect to $M$, we ensure that it will not waste capacity on spurious order-specific features. The inductive bias for this type of task differs from other settings like machine translation where word order is meaningful.

## B.1 Definitions and Lemmas

A permutation equivariant function is one that achieves the same result if you permute the set then apply the function to the permuted set, as you would achieve if you had applied the function to the set then applied the same permutation to the output set. We provide a formal description in Def B.1.

**Definition B.1** (Permutation Equivariance). *Let $\mathbb{X}$ be the set of $n$ vectors $\{\boldsymbol{x}_1, \ldots, \boldsymbol{x}_n\}$, and $S_n$ the group of permutations of $n$ elements $\pi : \mathbb{X} \to \mathbb{X}$. Suppose we have a function $f : \mathbb{X} \to \mathbb{X}$, we call $f$* ***Permutation Equivariant*** *iff $f(\pi(\mathbb{X})) = \pi(f(\mathbb{X}))$.*

One of the central components of the Sequential Set Transformer is the Multi-Head Self Attention function (MHSA : $\mathbb{X} \to \mathbb{X}$) originally introduced in (Vaswani et al., 2017). MHSA computes an updated representation for one input vector in a set of input vectors by processing its relation with a collection of other vectors. Without the explicit addition of positional information to the content of these vectors, this operation is *invariant* to input order. MHSA is a content-based attention mechanism, and updates the contents of a *single* input vector based on the contents of the others. However, if we define $\mathbf{f}_{\text{MHSA}}$ as the function applying MHSA to update *each* input vector, then we have created a permutation equivariant function, described in Lemma B.1.

**Lemma B.1** ($\mathbf{f}_{\text{MAB}}$ is Permutation Equivariant on $M$). *Let $\mathbb{X}$ be a set of $D$-dimensional vectors $\{\boldsymbol{x}_1, \ldots, \boldsymbol{x}_M\}$, and $\mathbf{f}_{\text{MAB}} : \mathbb{R}^{D \times M} \to \mathbb{R}^{D \times M}$ be a function that applies MAB (as defined in Equation 1 of the main text) to each $\boldsymbol{x}_m \in \mathbb{X}$. Then $\mathbf{f}_{\text{MAB}}(\mathbb{X}_\tau)$ is permutation equivariant because $\forall \pi \in S_m$, $\mathbf{f}_{\text{MAB}}(\pi(\mathbb{X})) = \pi(\mathbf{f}_{\text{MAB}}(\mathbb{X}))$.*

Having discussed the preliminaries, we now show the permutation equivariance of AutoBot's encoder.

## B.2 AutoBot Encoder Properties

**Theorem B.2** (AutoBot Encoder is Permutation Equivariant on $M$). *Let $\mathbf{X}$ be a tensor of dimension $(K, M, t)$ where $K, M, t \in \mathbb{N}$, and where selecting on $t$ retrieves an invertible multiset of sequences, denoted $\mathbb{X}_\tau = \{\boldsymbol{x}_1, \ldots, \boldsymbol{x}_M\}$, where $\tau \in \{1, \ldots, t\}$ and $\boldsymbol{x}_m \in \mathbb{X}_\tau$ are $K$-dimensional. The AutoBot encoder, as defined in Section 3.1, is permutation equivariant on $M$.*

*Proof.* What follows is a direct proof of the permutation equivariance of the AutoBot encoder. The encoder is a composition of three functions, rFFN: $\mathbb{R}^{K \times t} \to \mathbb{R}^{d_K \times t}$, $\mathbf{f}_{\text{MAB}} : \mathbb{R}^{d_k \times M} \to \mathbb{R}^{d_k \times M}$, and $\mathbf{f}_{\text{MAB}} : \mathbb{R}^{d_K \times t \times M} \to \mathbb{R}^{d_K \times t \times M}$.

First, we show that the embedding function rFFN is equivariant on $M$. Let rFFN represent the application of a function $e \colon \mathbb{R}^K \to \mathbb{R}^{d_K}$ to each element of a set $\mathbb{X}_\tau$ where $\tau \in \{1, \ldots, t\}$. Then rFFN is equivariant with respect to $M$ by Definition B.1. Specifically, we can see that the function rFFN satisfies the equation $\forall \pi \in S_M$, rFFN$(\pi(\mathbb{X}_\tau)) = \pi(\text{rFFN}(\mathbb{X}_\tau))$. In the AutoBot encoder, the

function rFFN is applied to each set $\mathbb{X}_\tau$, and because it is permutation equivariant on $M$ for each, the application to each set represents a permutation equivariant transformation of the entire tensor.

Next, we show that the function $\mathbf{f}_{\text{MAB}}$, which applies a Multi-Head Attention Block (MAB) to each set, $(\mathbb{X}_1, \ldots, \mathbb{X}_t)$, is equivariant on $M$. $\mathbf{f}_{\text{MAB}}$ is permutation equivariant with respect to the $M$ dimension of $\mathbf{X}$ because MAB is permutation equivariant on each set $\mathbb{X}_\tau$ by Lemma B.1.

Finally, we define a function $\mathbf{f}_{\text{MAB}}$ which applies a positional encoding along the time dimension of $\mathbf{X}$ then a Multi-head Attention Block (MAB) to each matrix in $\mathbf{X}$, $\{\mathbf{X}_1, \ldots, \mathbf{X}_M\}$. $\mathbf{f}_{\text{MAB}}$ is permutation equivariant with respect to the $M$ dimension of $\mathbf{X}$ because the collection of matrices $\{\mathbf{X}_1, \ldots, \mathbf{X}_M\}$ is an unordered set, and the uniform application of a function to each element of this set satisfies permutation equvariance.

A composition of equivariant functions is equivariant, so the encoder is equivariant on $M$. □

## B.3 AUTOBOT DECODER PROPERTIES

We now provide a direct proof the the permutation equivariance of the AutoBot decoder on $M$. The decoder is an auto-regressive function initialized with a seed set $\mathbb{X}_t$ containing $M$ vectors of dimension $d_k$, each concatenated with a vector of latent variables and other context of dimension $2d_k$. For $\tau \in \{1, \ldots, T-t\}$ iterations, we concatenate the output of the decoder with the seed, the previous decoder output, and the latents to produce a tensor of dimension $(3d_K, M, \tau)$. In particular, the decoder is a composition of three functions, $\text{rFFN}_{dec}: \mathbb{R}^{3d_K \times \tau} \to \mathbb{R}^{d_K \times \tau}$, $\mathbf{f}_{\text{MAB}}: \mathbb{R}^{d_k \times M} \to \mathbb{R}^{d_k \times M}$, and $\mathbf{f}_{\text{MABD}}: \mathbb{R}^{d_K \times M \times \tau} \to \mathbb{R}^{d_K \times M \times \tau}$.

**Theorem B.3** (AutoBot Decoder is Permutation Equivariant on $M$). *Given an invertible multiset $\mathbb{X}_t$ with $M$ vector-valued elements, and an auto-regressive generator iterated for $\tau \in \{1, \ldots, T-t\}$, the AutoBot decoder, formed by a composition of three functions, $\text{rFFN}_{dec}: \mathbb{R}^{3d_K \times M} \to \mathbb{R}^{d_K \times M}$, $\mathbf{f}_{\text{MAB}}: \mathbb{R}^{d_k \times M} \to \mathbb{R}^{d_k \times M}$, and $\mathbf{f}_{\text{MABD}}: \mathbb{R}^{d_K \times \tau \times M} \to \mathbb{R}^{d_K \times \tau \times M}$, is equivariant on $M$.*

*Proof.* First, we must establish that the function $\text{rFFN}_{dec}: \mathbb{R}^{3d_K \times M} \to \mathbb{R}^{d_K \times M}$ is equivariant with respect to $M$. The AutoBot decoder applies $\text{rFFN}_{dec}$ to each set $\mathbb{X}_\tau$ where $\tau \in \{1, \ldots, T-t\}$, transforming the $M$ vectors of dimension $3d_K \to d_k$. Because $\text{rFFN}_{dec}$ represents a uniform application of $e_{dec}$ to each element of the set, we can see that it satisfies the definition of permutation equivariance, specifically that $\forall \pi \in S_M$, $\text{rFFN}_{dec}(\pi(\mathbb{X}_\tau)) = \pi(\text{rFFN}_{dec}(\mathbb{X}_\tau))$.

Next, we take the function $\mathbf{f}_{\text{MABD}}$, which first applies a function $\text{PE}: \mathbb{R}^{d_k, M, \tau} \to \mathbb{R}^{d_k, M, \tau}$ to $\mathbf{X}$ adding position information along the temporal dimension, then applies $f_{\text{MABD}}: \mathbb{R}^{d_K \times \tau} \to \mathbb{R}^{d_K \times \tau}$ to each matrix $\{\mathbf{X}_1, \ldots, \mathbf{X}_M\}$. $\mathbf{f}_{\text{MABD}}$ is permutation equivariant with respect to the $M$ dimension of $\mathbf{X}$ because the collection of matrices $\{\mathbf{X}_1, \ldots, \mathbf{X}_M\}$ is an unordered set, and the uniform application of a function to transform each element independently does not impose an order on this set.

Similar to the final step of the previous proof, we see that a function $\mathbf{f}_{\text{MAB}}$ applying a multi-head self-attention block (MAB) to each set $\{\mathbb{X}_{t+1}, \ldots, \mathbb{X}_\tau\}$ is equivariant on $M$ because MAB is permutation equivariant on each set (see Lemma B.1).

The composition of equivariant functions is equivariant, so the decoder is equivariant. □

## C MODEL DETAILS

### C.1 EGO-CENTRIC FORECASTING

AutoBots can be used to predict the future trajectories of all agents in a scene. However, in some settings we may wish to forecast the future trajectory of only a single agent in the set. We call this variant "AutoBot-Ego". Referring back to the encoder presented in Section 3.1, instead of propagating the encoding of all elements in the set, we can instead select the state of that element produced by the intra-set attention, $\boldsymbol{s} \in \mathbb{S}_\tau$. With this, our encoder proceeds with the rest of the computation identically to produce the tensor $\mathbf{C}_{1:t}^m$, which is an encoding of the ego element's history conditioned on the past sequence of all elements.

AutoBot-Ego's decoder proceeds as presented in section 3.2, with one exception. Since AutoBot-Ego only generates the future trajectory of an ego element, we do not perform intra-set attention in the

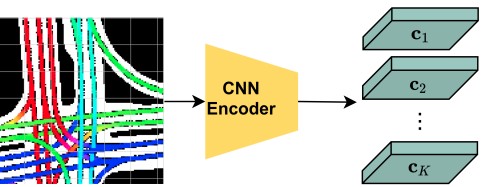

Figure 5: **AutoBots Context Encoder.** Example of the birds-eye-view road network provided by the Nuscenes dataset. Our CNN encoder produces a volume which we partition equally into $K$ modes. This allows each generated trajectory to be conditioned on a different representation of the input image.

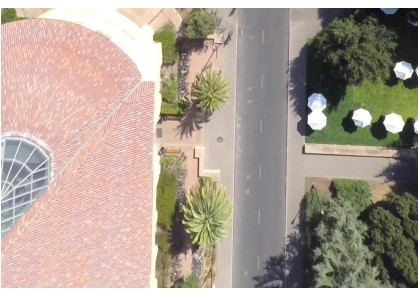

Figure 6: **TrajNet Map Example.** Example birds-eye-view map provided by the TrajNet benchmark. The footage was captured by a drone and pedestrians and bicycles are freely moving through the scene.

decoder. As a consequence of only dealing with a single ego-agent in the future generation, we do not repeat the seed parameter matrices $\mathbf{Q}_i$ across the $M$ dimension, where $i \in \{1, \ldots, c\}$. The objective function is updated to compute the likelihood of one element's future given the entire input sequence. That is, in AutoBot-Ego, $\mathcal{Y} = \mathbf{X}_{t+1:T}^m$.

## C.2 EXPECTATION MAXIMIZATION DETAILS

Our model also computes the distribution $P(Z|\mathbf{X}_{1:t})$ of discrete random variables. To achieve this, we employ $c$ learnable vectors (one for each mode) concatenated into the matrix $\mathbf{P} \in \mathbb{R}^{(d_K,c)}$, which behave like seeds to compute distribution over modes prior to observing the future (these seed parameters should not be confused with the decoder seed parameters $\mathbf{Q}$). The distribution is generated by performing the computations

$$p(Z|\mathbf{X}_{1:t}) = \text{softmax}(\text{rLin}(\mathbf{F})), \quad \text{where} \quad \mathbf{F} = \text{MABD}(\mathbf{P}_{1:c}, \mathbf{C}),$$

where rLin is a row-wise linear projection layer to a vector of size $c$.

## C.3 CONTEXT ENCODING DETAILS

In the nuScenes dataset, we are provided with a birds-eye-view $128 \times 128$ RGB image of the road network, as shown in Figure 5. In the TrajNet dataset, we are also provided with a birds-eye-view of the scene, as shown in Figure 6. We process this additional context using a 4 layer convolutional neural network (CNN) which encodes the map information into a volume of size $7 \times 7 \times (7 * c)$ where $c$ is the number of modes. We apply a 2D dropout layer with a rate of $0.1$ on this output volume before processing it further. As our model employs discrete latent variables, we found it helpful to divide this volume equally among all $c$ modes, where each mode receives a flattened version of the $7 \times 7 \times 7$ volume. As described in Section 3.2, this context is copied across the $M$ and $T$ dimensions, and concatenated with the decoder seed parameters during the sequence generation process. Intuitively, each mode's generated trajectory is conditioned on a different representation of the context.

## C.4 IMPLEMENTATION AND TRAINING DETAILS

We implemented our model using the Pytorch open-source framework. Table 4 shows the values of parameters used in our experiments across the different datasets. The MAB encoder and decoder blocks in all parts of AutoBots use dropout. We train the model using the Adam optimizer with an initial learning rate. For the Nuscenes and TrajNet++ datasets, we anneal the learning rate every 10 epochs for the first 20 epochs by a factor of 2, followed by annealing it by a factor of $1.33$ every 10 epochs for the next 30 epochs. For the Argoverse dataset, we anneal the learning rate every 5 epochs by a factor 2 for the first 20 epochs. In all datasets, we found it helpful to clip the gradients

| Parameter | Description | Toy | OmniGlot | Nuscenes | Argoverse | TrajNet | TrajNet++ |
|---|---|---|---|---|---|---|---|
| $d_k$ | Hidden dimension throughout all parts of the model. | 64 | 128 | 128 | 128 | 128 | 128 |
| Learning Rate | Learning rate used in Adam Optimizer. | 1e-4 | 5e-4 | 7.5e-4 | 7.5e-4 | 7.5e-4 | 5e-4 |
| Batch size | Batch size used during training. | 6 | 64 | 64 | 64 | 64 | 64 |
| c | Number of discrete latent variables. | 10 | 4 | 10 | 6 | 5 | 6 |
| $\lambda_e$ | Entropy regularization strength. | 3.0 | 1.0 | 30.0 | 30.0 | 5.0 | 30.0 |
| Dropout | Amount of dropout used in MHSA functions. | 0.0 | 0.2 | 0.1 | 0.1 | 0.3 | 0.05 |
| $L_{enc}$ | Number of stacked social/Temporal blocks in the encoder. | 1 | 1 | 2 | 2 | 2 | 2 |
| $L_{dec}$ | Number of stacked social/Temporal blocks in the decoder. | 1 | 1 | 2 | 2 | 2 | 2 |
| H | Number of attention heads in all MHSAs. | 8 | 8 | 16 | 16 | 8 | 16 |

Table 4: Hyperparameters used for AutoBots across all four datasets.

to a maximum magnitude of 5.0. For the Nuscenes experiments, our model takes approximately 80 epochs to converge, which corresponds to approximately 3 hours of compute time on a single Nvidia Geforce GTX 1080Ti GPU, using approximately 2 GB of VRAM. We used mirroring of all trajectories and the map information as a data augmentation method since the dataset's scenes originate from Singapore and Boston. For Argoverse experiments, our model takes approximately 30 epochs to converge, which corresponds to approximately 10 hours on the same resources. In order to improve the performance, we use other agents in the certain scenes as an ego-agent, effectively doubling the size of the dataset. We chose agents that move by at least 10 meters. This, in-turn, increases the training time to approximately 24 hours. For the TrajNet, TrajNet++ and Omniglot datasets, our model takes approximately 100 epochs which corresponds to a maximum 2 hours of compute time on the same resources.

### C.5 PERFORMANCE - COMPARING SEED PARAMETER DECODER WITH AUTOREGRESSIVE DECODER

One of the main architectural contributions of our work compared to our contemporaries is the use of seed parameters in the decoder network to allow for one-shot prediction of the entire future trajectory (compared to the established way of rolling out the model autoregressively timestep by timestep). To this end, we created an ablation of our model that predicts the future timesteps autoregressively and we compare its performance to ours across different variables (number of agents, input sequence length, output sequence length, and number of modes). Figure 7 shows the inference rate (i.e. full trajectory predictions per second) of AutoBot-Ego (top) and AutoBot (bottom). We can see that in the single-agent case (AutoBot-Ego ), both the seed parameter version of the model as well as the autoregressive ablation are able to operate in real-time ($\geq$ 30 FPS), save for when using a prediction horizon longer than 20 steps. We chose 30 FPS as a real-time indicator line since most cameras used in self-driving cars output at least 30 FPS (Yeong et al., 2021); Lidar data is usually slower to retrieve at 5-20 FPS. The performance difference is more obvious when comparing AutoBot to its ablation. The autoregressive model is barely able to perform at 30 FPS when the number of agents is low, the number of input steps is very low, the predicted future steps are under 20, and the number of modes is low.

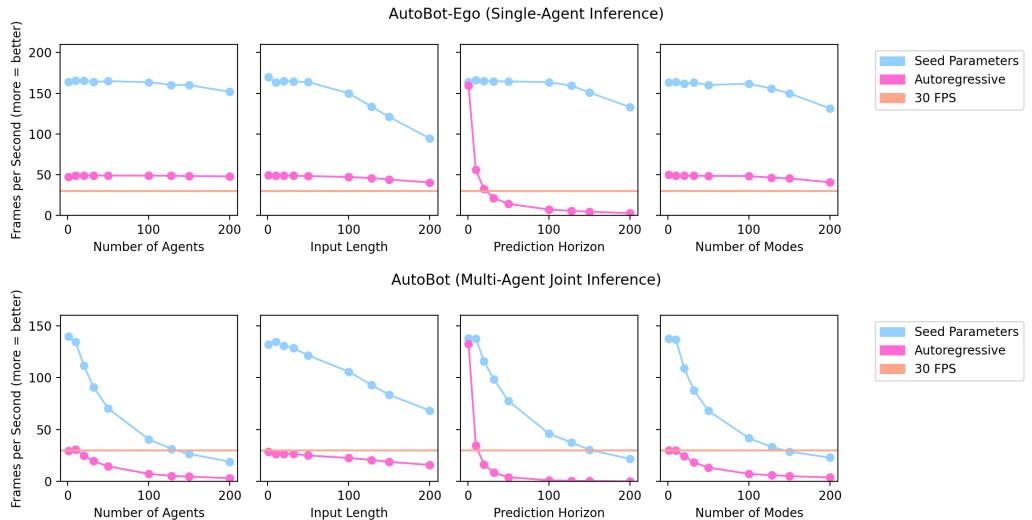

Figure 7: **AutoBots's Inference Rate** To show the impact of using a seed parameter decoder as opposed to an autoregressive decoder, we compare the inference rate of AutoBots with its autoregressive but otherwise identical ablation. We plot how the inference rate varies with increasing size of input/output variables. From left-to-right, these variables correspond to 1) number of agents, 2) number of observable timesteps for each agent, 3) prediction horizon, and 4) number of modes. The top row corresponds to AutoBot-Ego while the bottom row corresponds to AutoBot.

# D  AUTOBOTS ADDITIONAL RESULTS

## D.1  ABLATION STUDY ON NUSCENES

Table 5 shows the results of ablating various components from AutoBot-Ego on the Nuscenes dataset. As we can see, the model trained without the map information performs significantly worse across all metrics. This shows that indeed AutoBot-Ego learns to attend to the map information effectively. The 'No Latent Variable' version of our model is a model without the latent variable, and therefore, without the decoder seed parameters. This model then becomes auto-regressive and samples from the output bivariate Gaussian distribution at every timestep. We note that this version of the model is significantly slower to train (taking approximately three times as long) due to its autoregressive nature. In order to generate ten trajectories, the generation process is repeated $c = 10$ times. This model uses the same transformer encoder and decoder blocks as described in Section 3. As we can see from its performance on the ADE metrics, without latent variables, the model is not capable of generating diverse enough trajectories. This is expected since the output model $\phi$ is now tasked with accounting for both the aleatoric and epistemic uncertainties in the process. Finally, the 'No Data Augmentation' model shows that although it is helpful to augment the data with mirroring, even without it the model is still capable of producing strong performance.

| AutoBot-Ego | Min ADE (5) | Min ADE (10) | Miss Rate Top-5 (2m) | Miss Rate Top-10 (2m) | Off Road Rate |
|---|---|---|---|---|---|
| No Map Information | 1.88 | 1.35 | 0.72 | 0.58 | 0.22 |
| No Latent Variable | 1.77 | 1.34 | 0.76 | 0.64 | 0.07 |
| No Data Augmentation | 1.47 | 1.08 | **0.66** | 0.47 | **0.03** |
| AutoBot-Ego | **1.43** | **1.05** | **0.66** | **0.45** | **0.03** |

Table 5: **NuScenes Ablation Study.**. We study the performance benefits of AutoBot-Egoover various ablated counterparts.

## D.2 ANALYSIS OF ENTROPY REGULARIZATION

This dataset contains the trajectories of a single particle having identical pasts but diverse futures, with different turning rates and speeds, as shown in Figure 8 (top row). In the second row of Figure 8, we show that our model trained without entropy regularization, $\lambda_e = 0.0$, can cover all modes. However, the short trajectory variants are all represented using only one of the modes (2nd row, 3rd column), as shown by the growth of the uncertainty ellipses as the trajectory moves forward. This shows that AutoBots learn to increase the variance of the bivariate Gaussian distribution in order to cover the space of trajectories. The third row of Figure 8, shows the learned trajectories for this variant trained with $\lambda_e = 3.0$. As we can see, the entropy regularization penalizes the model if the uncertainty of bivariate Gaussians is too large. By pushing the variance of the output distribution to a low magnitude, we restrict the model such that the only way it can achieve high likelihood is to minimize the distance between the mean of the bivariate distributions and ground-truth trajectories.

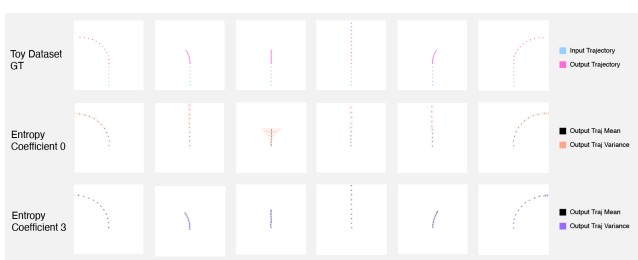

Figure 8: **Influence of entropy loss on particle experiment.** The input trajectory (cyan, only shown in top row) is identical in all cases. The model trained without an entropy loss term (middle row) covers all modes with high variance, while the model with moderate entropy regularization (bottom row) is able to learn the modes with low variance and without overlap.

**Ablation Study - Importance of Entropy Loss Term**. In Table 6, we show an ablation study to evaluate the importance of the entropy component on training the model for nuScenes. Again, we observe that for multi-modal predictions, enforcing low entropy on the predicted bivariate distributions has a great positive impact on the results, with an increased $\lambda_e$ resulting in better performance. This was observed on all evaluated metrics with the exception of the min FDE, which was poor regardless of the model. Furthermore, we can see from Table 1 that, in general, all prior models have a difficulty obtaining strong performance on the min FDE (1) metric, with the best value having an overall error of $8.49$ meters. This may be due to the difficulty of predicting the correct mode when one only has access to two seconds of the past sequence.

| $\lambda_e$ | Min ADE (5) | Min ADE (10) | Miss Rate Top-5 (2m) | Miss Rate Top-10 (2m) | Min FDE (1) | Off Road Rate |
|---|---|---|---|---|---|---|
| 0 | 1.75 | 1.29 | 0.67 | 0.58 | 9.55 | 0.10 |
| 1 | 1.71 | 1.22 | 0.65 | 0.57 | 9.03 | 0.08 |
| 5 | 1.70 | 1.14 | 0.62 | 0.53 | 9.11 | 0.05 |
| 30 | 1.75 | 1.14 | 0.63 | 0.54 | 9.51 | 0.04 |
| 40 | 1.72 | 1.11 | 0.60 | 0.51 | 9.01 | 0.04 |

Table 6: **NuScenes Quantitative Results by Entropy Regularization**. We study AutoBot-Ego's performance when varying the entropy regularization term on nuScenes, and see that increased entropy improves model performance (particularly Min ADE 10 and Off Road Rate) up to a limit.

## D.3 TRAJNET++ ABLATION AND QUALITATIVE RESULTS

Table 7 shows the effect of increasing the number of social/temporal encoder and decoder layers in the general AutoBot architecture. We can see that increasing the number of layers in the decoder has a positive impact on the performance of the joint metrics. This is expected since as we increase the number of social layers, the model can use high-order interactions to model the scene. Figure 9 shows an example scene generated by each of the model variants.

## D.4 TRAJNET, A PEDESTRIAN MOTION PREDICTION DATASET

We further validate AutoBots on a pedestrian trajectory prediction dataset TrajNet (Sadeghian et al., 2018) where we are provided with the state of all agents for the past eight timesteps and are tasked

| Model | Ego Agent's Min ADE(6) (↓) | Number Of Collisions (↓) | Scene-level Min ADE(6) (↓) | Scene-level Min FDE(6) (↓) |
|---|---|---|---|---|
| $L_{enc} = 1, L_{dec} = 1$ | 0.108 | 273 | 0.152 | 0.281 |
| $L_{enc} = 2, L_{dec} = 1$ | 0.102 | 214 | 0.138 | 0.254 |
| $L_{enc} = 3, L_{dec} = 1$ | 0.099 | 227 | 0.135 | 0.245 |
| $L_{enc} = 1, L_{dec} = 2$ | 0.099 | 164 | 0.132 | 0.248 |
| $L_{enc} = 1, L_{dec} = 3$ | 0.091 | 123 | 0.121 | 0.225 |
| $L_{enc} = 2, L_{dec} = 2$ | 0.095 | 139 | 0.128 | 0.234 |
| $L_{enc} = 3, L_{dec} = 3$ | 0.090 | 98 | 0.119 | 0.216 |

Table 7: TrajNet++ ablation studies for a multi-agent forecasting scenario. We investigate the impact of the number of social/temporal layers in the encoder and the decoder.

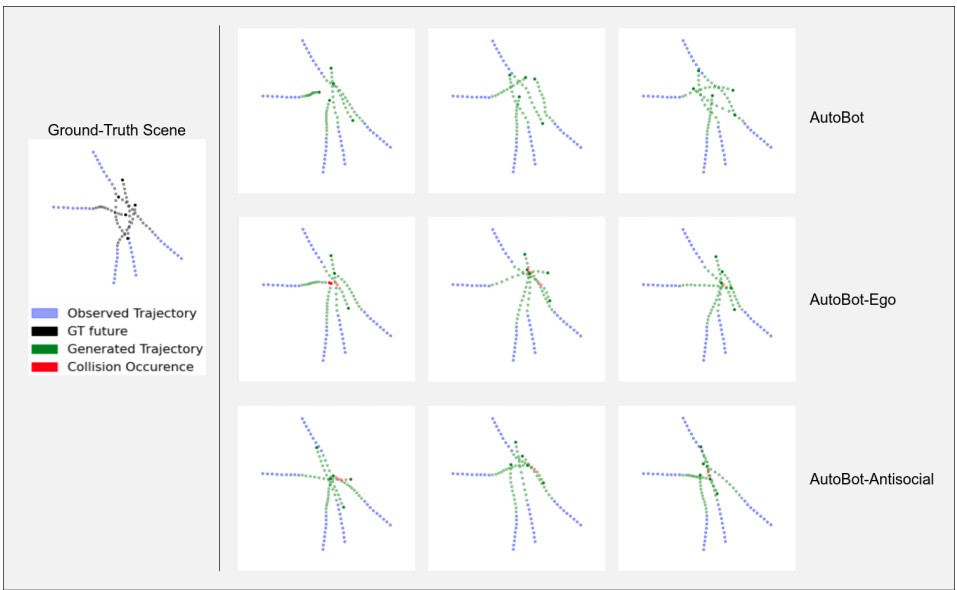

Figure 9: **TrajNet++ qualitative results.** In this example scene, we see five agents moving towards each other with their input trajectories (blue) and their ground-truth future trajectories (black) shown in the left figure. The different rows show three of the $c = 6$ scene predictions made by each model variant. AutoBot-Ego and AutoBot-AntiSocial produce some modes containing collisions or socially inconsistent trajectories, while AutoBot, which has social attention in the decoder, achieves consistent trajectories.

with predicting the next 12 timesteps. In this experiment, we apply our general architecture (AutoBot) to a multi-agent scene forecast situation, and ablate various the social components in the decoder only, and decoder and encoder.

| Model | Number Of Collisions | Scene-level Min ADE(5) | Scene-level Min FDE(5) |
|---|---|---|---|
| AutoBot-AntiSocial | 474 | 0.571 | 1.02 |
| AutoBot-Ego | 466 | 0.602 | 1.07 |
| AutoBot | **326** | **0.471** | **0.833** |

Table 8: **TrajNet ablation studies for a multi-agent forecasting scenario.** We evaluate the impact of using the extra social data as a set and the actual multi-agent modelling on this context. We found that AutoBot is able to cause fewer collisions between agents compared to variants without the social attention.

In Table 8, we present three variants of the model. The first, AutoBot-AntiSocial corresponds to a model without the intra-set attention functions in the encoder and decoder (see Figure 2). The second variant, AutoBot-Ego, only sees the past social context and has no social attention during

the trajectory generation process. The third model corresponds to the general architecture, AutoBot. All models were trained with $c = 5$ and we report scene-level error metrics as defined by (Casas et al., 2020). Using the social data with the multi agent modelling in the decoder greatly reduces the number of collisions between predicted agent futures. Furthermore, we can see that AutoBot significantly outperforms its ablated counterparts on generating scenes that match ground-truth, as observed by the reduced scene-level minADE and minFDE. We expect this theoretically since the ego-centric formulation makes the independence assumption that the scene's future evolution can be decomposed as the product of the individual agent's future motion in isolation, as remarked in (Casas et al., 2020). AutoBot does not make this assumption as we condition between agents at every timestep.

Figure 10 shows an example of multi-agent trajectory generation using the three different approaches for predicting the scene evolution. We present four of the five modes generated by the model for this example input scene. One can observe that the only model achieving scene-consistent predictions across all modes is AutoBot . Interestingly, we also observe that the different modes correspond to different directions of motion, highlighting the utility of using modes. In fact, the modes in AutoBot condition the entire scene future evolution, producing alternative realistic multi-agent futures. We refer the reader to Figures 11, 12, 13 and 14 where we show some additional qualitative results of AutoBots' variants compared to AutoBot on the TrajNet dataset. These results highlight the effectiveness of the sequential set attention mechanism in the decoding process of our method and how important it is to predict all agents in parallel compared to a 1-by-1 setup.

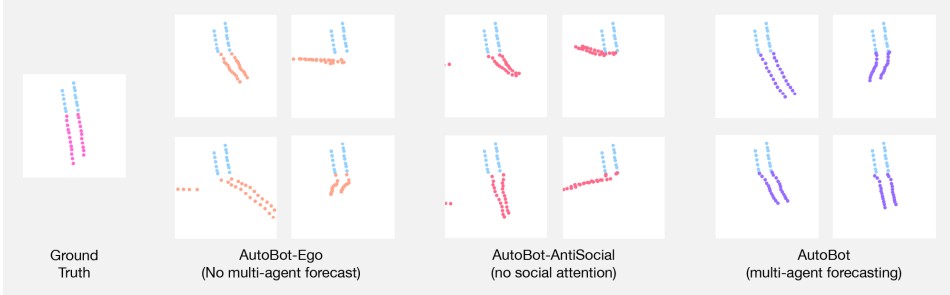

Figure 10: **TrajNet qualitative results.** In this example, we see two agents moving together in the past (cyan) and future (pink, left). We compare how different variants of AutoBots make predictions in this social situation. AutoBot-Ego and AutoBot-AntiSocial produce some modes containing collisions or unrealistic trajectories, while AutoBot, which has intra-set attention during decoding, achieves consistent trajectories.

### D.5 OMNIGLOT ADDITIONAL DETAILS AND RESULTS

The LSTM baseline used in our Omniglot experiments have a hidden size of 128 and use $c = 4$ latent modes. The input is first projected into this space using an embedding layer, which is identical to the one used in AutoBot. All input sequences of sets are encoded independently using a shared LSTM encoder. During the decoding phase, we autoregressively generate the next stroke step using an LSTM decoder. In order to ensure consistency between the predicted future strokes, inspired by Social LSTM (Alahi et al., 2016), the LSTM baseline employs an MHSA layer at every timestep operating on the hidden state of the different strokes making up the character. We concatenate a transformation of the one-hot vector representing the latent mode with the socially encoded hidden state at every timestep. The output model is identical to the one used AutoBot, generating a sequence of bivariate Gaussian distributions for each stroke. We performed a hyperparameter search on the learning rate and found the optimal learning rate to be $5e - 4$. Furthermore, as with all our other experiments, we found it helpful to employ gradient clipping during training.

We provide additional results on the two tasks defined in Section 4.4. Figure 15 shows additional successful AutoBot results on task 1 (completing multiple strokes) compared to an LSTM baseline equipped with social attention. These results highlight the effectiveness of sequential set transformers for generating consistent diverse futures given an input sequence of sets. Figure 16 shows examples where both models fail to correctly complete the character in task 1. Figure 17 compares AutoBot-Ego with the LSTM baseline on predicting a new stroke given the two first strokes of a character (task 2). These results highlight that although not all modes predict the correct character across all modes, all

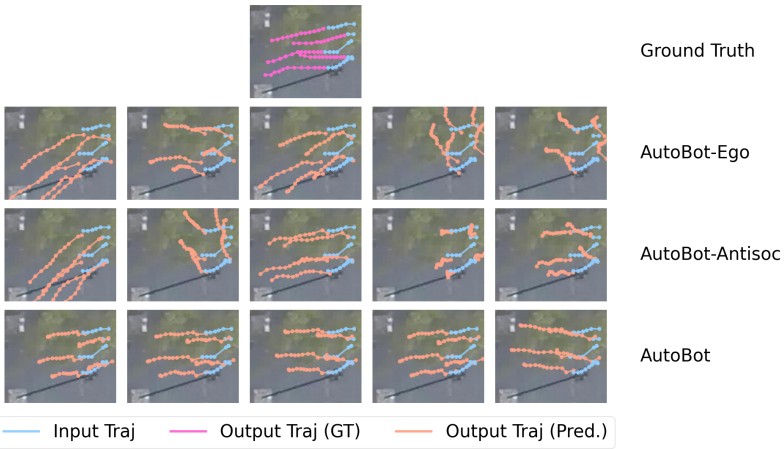

Figure 11: **TrajNet qualitative results 1/4.** Example scene with multiple agents moving together. These trajectories are plotted over the birds-eye-view image of the scene where we zoom into interesting trajectories. We can see that only AutoBot produces trajectories that are realistic in a group setting across all modes.

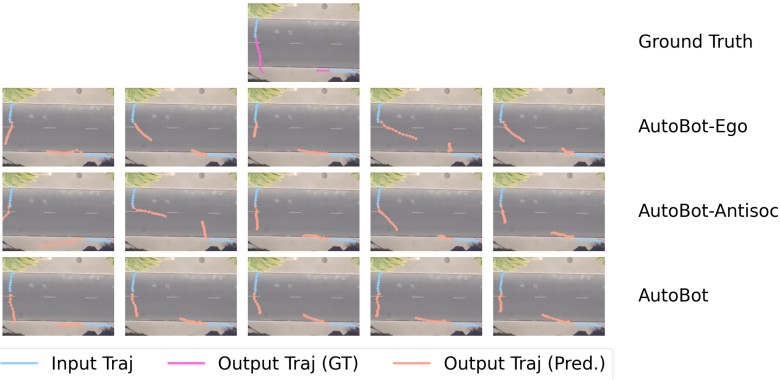

Figure 12: **TrajNet qualitative results 2/4.** Example scene with two agents moving separately in a road setting. We want to highlight this interesting scenario where some modes of AutoBots-Solo and AutoBots-AntiSocial results in trajectories that lead into the road, while AutoBot seems to produces trajectories more in line with the ground-truth, and lead the agent to cross the road safely.

generated predictions are consistent with realistic characters (i.e., confusing "F" with "Π" or "H" with "Π").

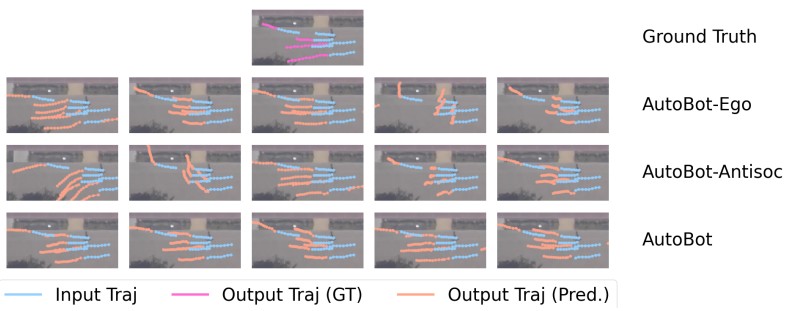

Figure 13: **TrajNet qualitative results 3/4.** Additional example scenes with multiple agents moving together. Again, we wish to highlight the advantage of modelling the scene jointly, which is evident by the results of AutoBot.

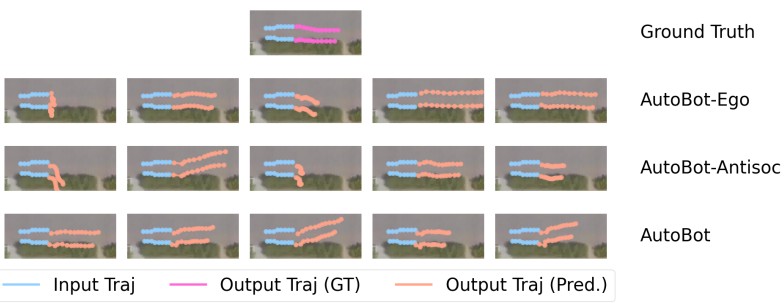

Figure 14: **TrajNet qualitative results 4/4.** Example scenes with two agents moving together. Again, we see that AutoBot produces trajectories consistent with the scene across all modes (e.g., not crashing into the bushes) and maintains the social distance between the walking agents.

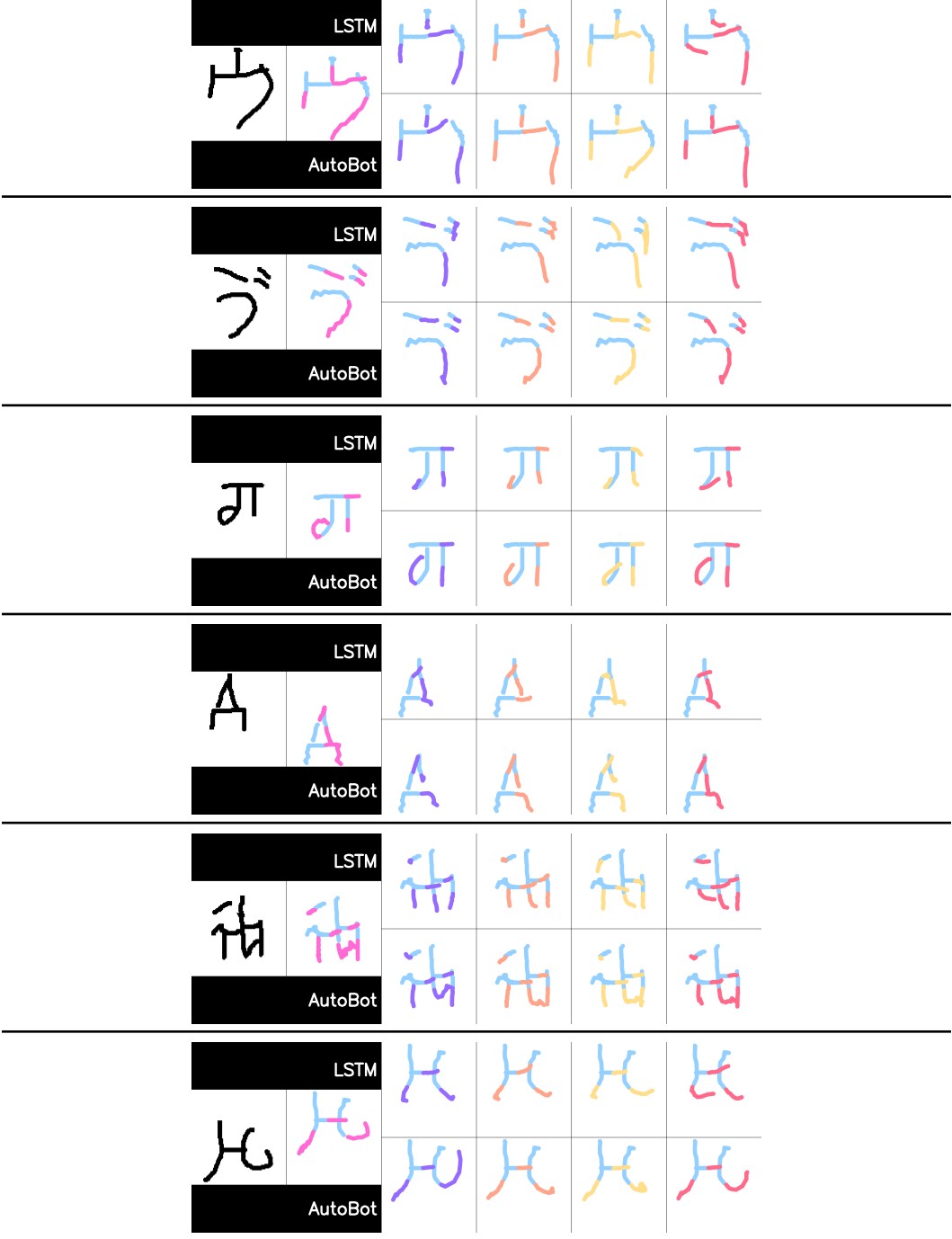

Figure 15: **Omniglot Task 1 Additional Results**. These are some additional random characters from the Omniglot stroke completion task. We can see that AutoBot produces plausible characters on the test set, while the LSTM struggles to create legible ones.

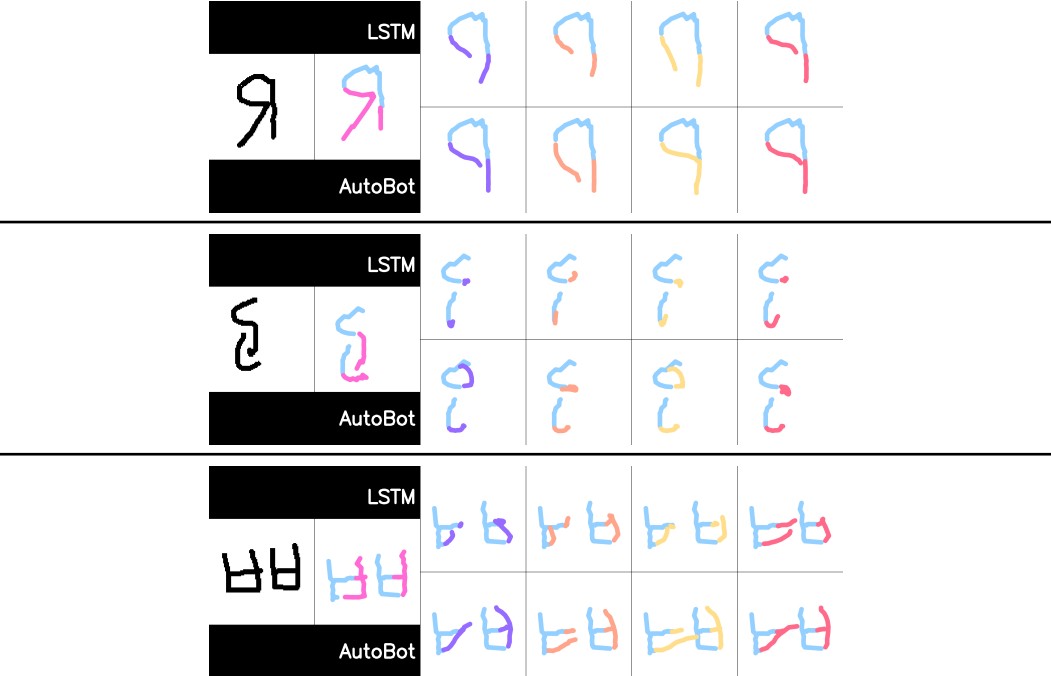

Figure 16: **Omniglot Task 1 Failure Cases**. There were some characters where AutoBot failed to learn the correct character completion. We can currently only speculate why that is the case. Our first intuition was that this might occur when there are 90 degree or less angles in the trajectory that is to be predicted but in Fig. 15, we can see that there are examples where this does not seem to be a problem. We believe that this might be due to a rarity of specific trajectories in the dataset but more investigation is required.

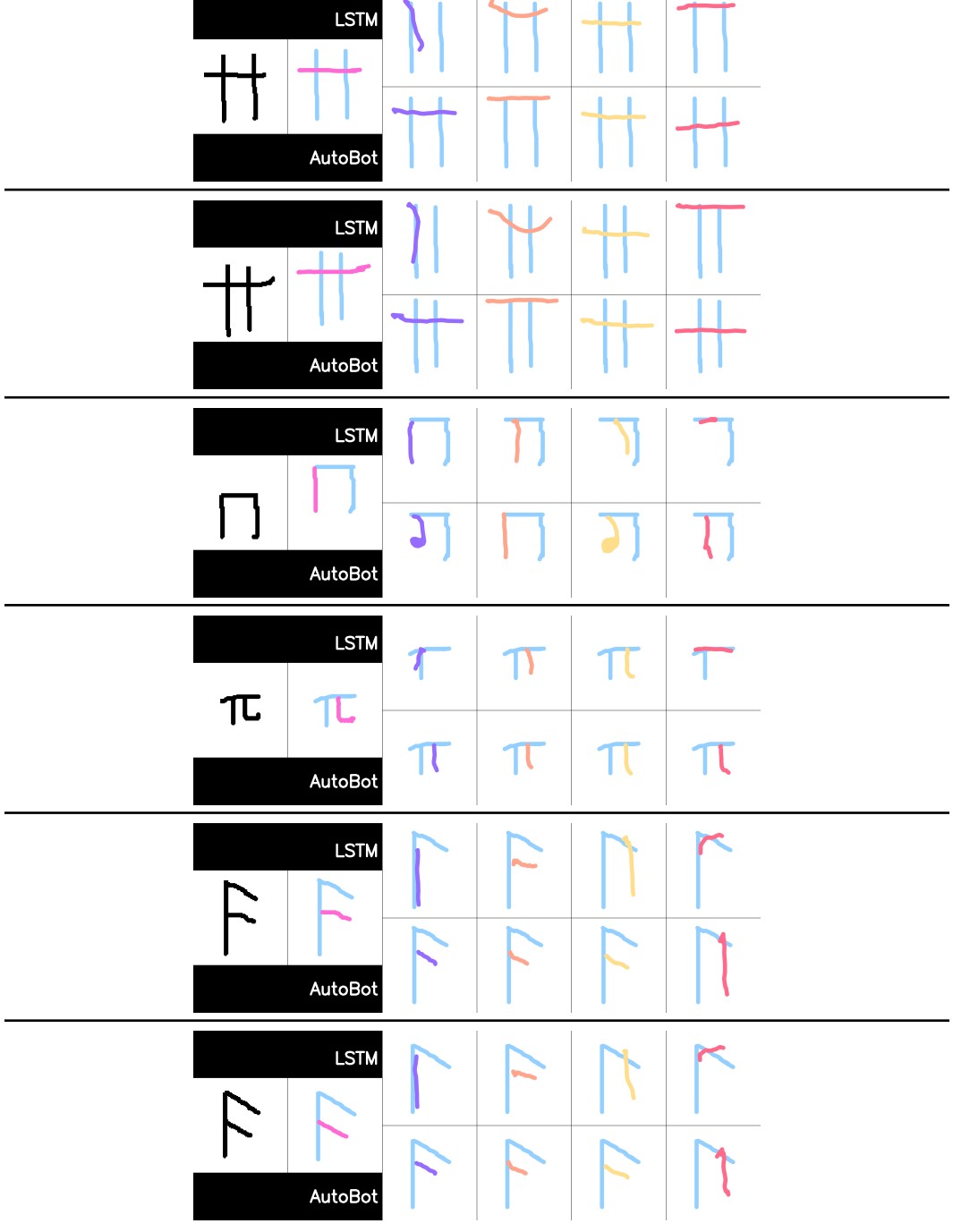

Figure 17: **Omniglot Task 2 Additional Results**. These are some additional random characters from the Omniglot character completion task. Again, we can see that AutoBot produces plausible characters on the test set, where different modes capture plausible variations (e.g. F to H and H to PI), while the LSTM struggles to capture valid characters in its discrete latents.

