# OpenReview forum: "Latent Variable Sequential Set Transformers for Joint Multi-Agent Motion Prediction"
_ICLR.cc/2022/Conference — ICLR 2022 Spotlight_

### Official Review · Reviewer_rcqg · 2021-11-02

**Correctness:** 4
**Technical Novelty And Significance:** 3
**Empirical Novelty And Significance:** 3
**Recommendation:** 8
**Confidence:** 4

**Details Of Ethics Concerns:**

This paper is similar to https://arxiv.org/pdf/2104.00563.pdf, if the arxiv one is not accepted by any other conferences yet and they are from the same people, I'm happy to accept this one. Otherwise the contribution/difference (i.e. minor change on the decoder) is very limited and should not be accepted.

**Main Review:**

Pros:
+ This proposed model is simple and effective. It applies multi-head attention modules on temporal and social axis both in encoder and decoder to capture motion and interaction among agents.
+ The proposed model has been tested with comprehensive experiments, including on toy examples, real-world dataset such as nuScenes, Argoverse and TrajNet, as well as an image generation dataset (Omniglot). It shows comparable results compared to previous state-of-the-art models.
+ The latent variable is used to represent the discrete future for each agents. In spirit, it is same as Multiple Future Prediciton (Tang et al 2019), however, this paper adapts it into the transformer model and make it work as well.
+ The training of this model is very fast, as pointed out on page 6, it consumes 0.4% computation of SceneTransformer and 5% of Jean's method on Argoverse. This shows the efficiency of the proposed model.
+ Code is provided and it's nice and clean.

Cons:
- It's related to Axial Attention in Multidimensional Transformers, Jonathan Ho et al, 2019, and should cite that work.
- In the provided code, the decoder is autoregressive instead of what is proposed in the main paper. It should be updated accordingly, and more ablation/discussion on these two type of decoder would be very helpful.
- Minor point: for equation (5) on L(\theta) = Q + const, the const should be KL divergence and is not a const w.r.t. \theta.


**Summary Of The Paper:**

This paper adapts transformer to multi-agent motion forecasting. The attention layers are applied on time and agent axis to capture motion and social information. A latent variable is introduced on the output to capture discrete motion for each agent.
Extensive experiments are conducted on various dataset with good performance. The training time of the proposed model is significantly faster than previous methods.

**Summary Of The Review:**

This paper compiles an interesting model for multi-agent motion forecasting. Although the transformer module is similar to Axial Transformer [Ho 2019] and the latent variable formulation and optimization are similar to MFP [Tang 2019], it is the first to successfully apply those two module together for motion forecasting task, with clean and concise code released. Thus I would vote for weak accept.

---

> ### Author Response · Authors · 2021-11-17
> **Response to Reviewer rcqg**
>
> We thank the reviewer for their valuable feedback on our manuscript. We are happy that the reviewer finds our model simple and effective, and that our model is very fast with training done at a fraction of the computational cost of other related approaches.
>
> *Relationship to Axial Attention in Multidimensional Transformers*
>
> We thank the reviewer for bringing this related work to our attention. This work proposes a way to apply attention to bidimensional data in a single axis without flattening. We apply self attention successively intercalating the social and the temporal axis. We will discuss and cite it accordingly.
>
> *The provided code is autoregressive.*
>
> Indeed, we unfortunately sent the code with autoregressive scene generation. We will be updating the code shortly with corrections. We will notify you when the supplementary material has been updated. Furthermore, we would like to point out that we plan to release our code to the public upon publication of our work.
>
> *Ablation studies and discussion on the autoregressive versus our approach would be helpful.*
>
> We would like to bring the reviewer’s attention to our general comment above where we presented multiple ablation studies. The second table shows an ablated version of Autobot where the model does not use any latent variables (i.e., seed parameters) and is autoregressive. We can clearly see that this model under-performs compared to our proposed seed parameters approach. In addition, auto-regressive sampling is computationally more expensive compared to the use of seed parameters.
>
> *Minor point on Equation 5: the const term should be the KL divergence and is not const wrt $\theta$.*
>
> Indeed, this term is constant with respect to $\theta$ since it corresponds to the entropy of the $q(z)=p_{\theta_{old}}(Z|\mathcal{Y},\mathbf{X}_{1:t})$
>
> which depends on $\theta_{old}$. We refer the reviewer to Section 9.4 (and specifically equation 9.74) of Bishop’s Pattern Recognition and Machine Learning (2006) book for the derivation of this term.
>
> *Regarding your final comment about an ArXiv paper:*
>
> We welcome any additional attention to this matter, we have respected all ICLR rules, and we encourage you to discuss your concern with reviewer **DomP** as it pertains to SceneTransformer and novelty while paying attention to the dates these papers were submitted to the ArXiv.

---

> > ### Author Response · Authors · 2021-11-24
> > **Code Update**
> >
> > Dear Reviewer rcqg,
> >
> > We wanted to inform you that we updated the Python notebook for the Omniglot dataset, which shows how Autobot employs the decoder seed parameters to generate characters.
> >
> > We believe the AC has seen your ethics comment and since this work was not desk rejected, and in light of our responses, we wanted to know if you would be willing to increase your score?
> >
> > Thanks,
> > The Authors

---

> > > ### Comment · Reviewer_rcqg · 2021-11-27
> > > **Thanks for the clarification and the updated code.**
> > >
> > > I've updated my score as well.

---

### Official Review · Reviewer_HbLp · 2021-11-02

**Correctness:** 3
**Technical Novelty And Significance:** 2
**Empirical Novelty And Significance:** 3
**Recommendation:** 6
**Confidence:** 4

**Main Review:**

Strengths: The paper proposes a solid architecture by considering the requirements of the problem. The proposed architecture relies on Transformers and the self-attention function which is an effective tool in modeling relations between different entities. Trajectory forecasting, especially in a multi-agent setting, is highly stochastic and may result in many feasible predictions. The paper incorporates a set of learnable seed parameters into the Transformer architecture to enable multi-moded prediction where the seeds are expected to lead to different modes. The model is evaluated in established benchmarks and achieves competitive performance with a lower computational requirement.

Weaknesses and Questions: The main contributions are the set assumption and the discrete latent variable. Unfortunately, the paper does not provide empirical evidence for these design choices. Ablations evaluating the proposed approaches could be insightful.

Could the authors clarify why the proposed set assumption is feasible for multi-agent tasks? I can see the advantage of this design choice if the set of agents is dynamic (i.e., gets updated in time). However, this scenario is not covered in the experimental setup. It would be useful to see if the set assumption contributes to the performance.

It wasn’t clear to me how the discrete latent variable is modeled. It is partly due to the cluttered notation (see the next comment). I kindly ask the authors to provide more details. An ablation evaluating the proposed model without using the discrete latent variable could be useful. I also would like to point out alternatives in case the authors are not aware of this line of work. Latent variables in sequence models have been studied [1]. While it is reported to be effective, there are more straightforward approaches to get multi-moded predictions [2]. Have the authors experimented with continuous latent variables or mixture density outputs for multi-modal predictions?

I find the notation not rigorous. Several symbols are used in multiple definitions, making the paper hard to follow. For example,

1- $Z$:
The discrete latent variable appears in the objective function in section 3.3 with the symbol $Z$. It is also defined to be the seed as “learnable seed parameters $Z \in \mathcal{R}^{(d_K; M; T)}$” in section 3.2. Are they the same?

2- $K$:
Section 3.1: “... having M elements (or agents) with K attributes (e.g. x/y position).”
Section 4.1: “... AutoBot-Ego trained with K = 10.” or “AutoBot-Ego obtains the best overall Min ADE out of 5 predictions (K = 5).”

I have the following questions:

3- Could the authors clarify this statement in section 4.1:  “In order to generate ten predictions from five modes, at test time, we create combinations between the latent modes one-hot vector by summing one-hot vectors.”
4- What is the difference between the “AutoBot-Ego (K=10)” and “AutoBot-Ego (ensemble)” entries in Tab. 1?

Finally, [3] seems to be relevant and should be included in the related work discussion if the authors also agree.

[1] Chung, Junyoung, et al. "A recurrent latent variable model for sequential data." Advances in neural information processing systems 28 (2015): 2980-2988.

[2] Graves, Alex. "Generating sequences with recurrent neural networks." arXiv preprint arXiv:1308.0850 (2013).

[3] Li, Jiachen, et al. "Evolvegraph: Multi-agent trajectory prediction with dynamic relational reasoning." arXiv preprint arXiv:2003.13924 (2020).

-- Post-rebuttal edit --
I thank the authors for the clarifications. I read other reviews and the responses. I am glad to see that the authors addressed the questions and provided additional experimental evidence. The ablations improve the quality of the submission. In fact, the latent variable ablation can be included in the main submission. Although the technical novelty is slightly limited, the proposed approach is solid. I increase my score to 6.

**Summary Of The Paper:**

This submission presents a Transformer-based architecture for trajectory prediction tasks involving multiple agents. Such a multi-agent setting requires learning of spatio-temporal representations capturing both the long-term temporal dependencies as well as the social interactions between the agents. The paper formulates the task as modeling of the sequence of sets where every entry in the set corresponds to an agent’s observation. The proposed architecture augments the set transformer with a discrete latent variable to be able to make multiple predictions into the future. The given seed sequence is first encoded into a context representation which is later used to make future predictions with a decoder where the agents can be modeled jointly or independent from each other.

**Summary Of The Review:**

The proposed architecture is tailored for the task of multi-agent trajectory prediction task and presented to be effective in the benchmarks. However, we do not know if it is due to the superior performance of the underlying computation units (i.e., transformer blocks) or the proposed extensions (i.e., discrete latent variable/learnable seed, set assumption). I also think that the presentation could be improved.

---

> ### Author Response · Authors · 2021-11-17
> **Response to Reviewer HbLp (1/2)**
>
> We thank the reviewer for their valuable feedback on our manuscript. We are happy that the reviewer appreciates that our architecture is solid and achieves competitive performance on established benchmarks at a fraction of the computational requirements of other methods.
>
> *Why is the set assumption feasible for multi-agent tasks?*
>
> The main purpose of the set formulation is that we believe it to be the correct way to formally talk and model such multi-agent settings. In practice, since we must deal with tensors for model computations, the dynamic number of agents in a scene is handled by employing masking in the softmax operation of the attention mechanism.
>
> Autobot encounters a dynamic number of agents in the Nuscenes dataset where, for certain scenes, we do not observe all four timesteps (2 seconds) of the input scene for all agents. The strong performance on Nuscenes validates that the attention masking approach works well in practice.
>
> In order to further study this part, we trained an Autobot model on the TrajNet++ synthetic dataset where we partially delete agent input trajectories (i.e., making the number of agents in the input scene change with time). The results of this model are presented in the table below:
>
> | Model | Ego minADE(c=6) | Number of Collisions | Scene-Level minADE (c=6) | Scene-Level minFDE (c=6) |
> | -------- | -------- | -------- | -------- | -------- |
> | AutoBot | 0.095 | 139 | 0.128 | 0.234 |
> | AutoBot (Dynamic number of agents) | 0.100 | 152 | 0.135 | 0.250 |
> *Numbers computed on 5451 validation scenes of TrajNet++ synthetic dataset.
> **The ego metric is computed on the agent around which the scene has been normalized.
>
> As we can see, the model is indeed capable of handling a dynamic number of agents with little impact on its performance. Furthermore, we are adding videos in the supplementary dataset labelled Autobot_with_dyn_num_agents. We hope that this has addressed your concerns over the set formulation.
>
>
> *How are the discrete latent variables modeled?*
>
> We would like to clarify how the discrete latent variables are modelled in Autobot. One of the main reasons that Autobot is faster than prior approaches is the use of the decoder seed parameters, which have a shape of $(d_k, T, c)$ (where $d_k$ is the model hidden size, $T$ is the prediction horizon, and $c$ is the number of modes being modelled). For joint prediction experiments (i.e., TrajNet++ or Omniglot task 1), this tensor is repeated across the agent dimension to produce a tensor of size $(d_k, T, M, c)$, where $M$ is the number of agents in the input scene.
>
> These seed parameters serve a dual purpose. The first is that they account for the diversity in future prediction, where each matrix $(d_k, T)$ corresponds to one setting of the discrete latent variable for a given agent. Secondly, they contribute to the speed of Autobot by allowing it to perform inference on the entire scene with a single forward pass through the decoder. Given each tensor, the decoder then deterministically generates the future scene. After doing this for all $c$ modes (efficiently done by folding it into the batch dimension), the decoder produces a tensor of size $(d_k, T, M, c)$ which can then be passed through the output model $\phi$ to produce the parameters of a bivariate gaussian distribution. We added a paragraph in Section 3.2 to provide the intuition on the decoder seed parameters.
>
> *Ablation without discrete latent variables would be useful.*
>
> Thank you for the suggestion of performing an ablation study on the discrete latent variable. For Autobot, ablating the discrete latent variable is equivalent to removing the decoder seed parameters. This means our model’s inference time increases due to the need to resort to auto-regressive sampling from the output model at every timestep. Nevertheless, we performed this ablation on the Nuscenes dataset, which is presented in our general comment above (labelled ‘No latent variable’).
>
> We can see that Autobot with the decoder seed parameters (i.e., discrete latent variables) performs significantly better than training without them. Furthermore, one of the issues with sampling from the bivariate gaussian is that we are not guaranteed to obtain the same results between runs on the same input scene. However, the main challenge for a model without latent variables is that the output model needs to incorporate both the aleatoric and epistemic uncertainty, which are both present in motion prediction problems.

---

> > ### Author Response · Authors · 2021-11-17
> > **Response to Reviewer HbLp (2/2)**
> >
> > *Have the authors experimented with continuous latent variables or mixture density outputs?*
> >
> > We have not experimented with continuous latent variables. One of the advantages of using discrete latent variables is that the model learns to generate the K most likely set of trajectories (or scenes) while maintaining the diversity between them. However, extending to the use of continuous latent variables constitutes an interesting extension of our proposed approach, which we hope to pursue in future work.
> >
> > The use of discrete latent variables is tightly connected to mixture density outputs. The difference is that in mixture density outputs, the multimodal output is on a per-timestep basis while our approach (which was also previously employed by MFP [1]) is multi-modal on the entire trajectory. The advantage of Autobot in modelling the discrete latent variables with seed parameters is that the model does not need to auto-regressively generate the future trajectory of the agent, and can instead generate the entire trajectory in a single forward pass.
> >
> > *Notation Clarification, mistakes and citation*
> >
> > Thank you for pointing out these overloaded symbols. We corrected these by renaming the seed parameter tensor to $Q$. We also changed  “$K=10$...” to “$c=10$...”, which is the symbol we used to represent the number of modes in section 3. We made multiple changes to section 3 to help clarify the presentation. We hope that the reviewer finds it clearer.
> >
> > Thank you for pointing out the statement in section 4.1, which is incorrect, we removed this sentence and corrected the section accordingly.
> >
> > Regarding the "Evolvegraph: Multi-agent trajectory prediction with dynamic relational reasoning”  paper, we would like to thank the reviewer for bringing this related work to our attention. We have added the citation in our related work section.
> >
> >
> > *We do not know if it is due to the superior performance of the underlying computation units (i.e., transformer blocks) or the proposed extensions (i.e., discrete latent variable/learnable seed, set assumption). I also think that the presentation could be improved.*
> >
> > We hope that the ablation studies we presented in the general comment have adequately addressed the reviewer's concerns and show that indeed the decoder seed parameters are an essential part of the model for both performance benefits and for computational efficiency. We note that the evaluations we performed with masked-out input agents validate that the set assumption is beneficial. Finally, we hope that the corrected notation has helped clarify the presentation of our work. Please see updated manuscript, specifically in section 3.
> >
> > [1] Tang, Y., & Salakhutdinov, R. (2019). Multiple Futures Prediction. NeurIPS.

---

> > > ### Author Response · Authors · 2021-11-26
> > > **Follow-up on Rebuttal**
> > >
> > > Dear Reviewer HbLp,
> > >
> > > Following your questions, we provided new ablation experiments that show the importance of the latent variable, and presented results on experiments where the number of agents in the scene dynamically changes. In addition, we improved the presentation of our method in Sections 3 and 4 following your recommendation.
> > >
> > > In light of this, we would like to obtain your current assessment of our work. Specifically, we would like to know whether you believe these additional experiments and updated exposition have addressed your concerns, and if so we hope that you would be willing to increase your score.
> > >
> > > Thank you for your time,
> > >
> > > The Authors

---

### Official Review · Reviewer_DomP · 2021-11-03

**Correctness:** 4
**Technical Novelty And Significance:** 2
**Empirical Novelty And Significance:** 2
**Recommendation:** 8
**Confidence:** 3

**Main Review:**

(+): Extensive experiments and results: The authors conducted experiments on 4 datasets, covering 3 different tasks (argo and nuscenes are both prediction for AV). These are very extensive. I do recognize that the method generalizes well to multiple different tasks.

(-) Novelty: I think the novelty should be good if it wasn't for the SceneTransformer paper. According to ICLR review guidelines, Arxiv papers published 30 days prior to the ICLR deadline are considered prior work. The SceneTransformer's formulation is very similar to that proposed in this paper, e.g. using transformers to model cross-agent and cross-time information propagation. Although SceneTransformer does not have an object query that is the latent variable Z, it was still able to learn a diverse set of future predictions. It's unclear if the explicit formulation of latent variable makes a difference. It seems that on Argoverse, this paper performs very similarly than SceneTransformer, which further confirms my view that the methods are very similar.

**Summary Of The Paper:**

The paper tackles the multi-agent trajectory prediction problem, primarily for autonomous driving, but experiments also include TrajNet and predicting Omniglot strokes. The authors claim their contributions in a very general sense:

- Novel method on modeling sequences of structured continuous variables and capture multi-modal distributions.
- Strong results on nuscenes, argoverse, trajnet and omniglot stroke prediction.

The key modeling novelty is the latent variable sequential set transformer which applies self-attention across different agents and across time in the scene.

I mostly agree with the authors' assessment in general. I have some concerns on the claimed novelty (see below).

**Summary Of The Review:**

I am giving a borderline reject for now due to the novelty concerns in light of a very similar paper in my opinion. I know that there are a lot of devils in the detail between the two works, and it would have been an obvious accept if this works outperforms SceneTransformer significantly.

In the rebuttal, may I ask the authors to further highlight the difference between their approach to SceneTransformer? And where do they see that their method is intuitively better than SceneTransformer?

POST REBUTTAL: The authors pointed out that SceneTransformer should be considered concurrent work. If that is the case, I have no issues to accept this paper.

---

> ### Comment · Program_Chairs · 2021-11-11
> **A clarification on the ICLR review guideline**
>
> Reviewer DomP noted in their review that "According to ICLR review guidelines, Arxiv papers published 30 days prior to the ICLR deadline are considered prior work."
>
> This is not correct based on the ICLR 2022 reviewer guide for the contemporaneous work ---
>
> https://iclr.cc/Conferences/2022/ReviewerGuide
>
> Q: Are authors expected to cite and compare with very recent work? What about non peer-reviewed (e.g., ArXiv) papers?
>
> A: We consider papers contemporaneous if they are published (available in online proceedings) within the last four months. That means, since our full paper deadline is October 5, if a paper was published (i.e., at a peer-reviewed venue) on or after June 5, 2021, authors are not required to compare their own work to that paper. Authors are encouraged to cite and discuss all relevant papers, but they may be excused for not knowing about papers not published in peer-reviewed conference proceedings or journals, which includes papers exclusively available on arXiv. Reviewers are encouraged to use their own good judgement and, if in doubt, discuss with their area chair.
>
> Given this, the reviewer DomP might need to revise their concerns about the relative novelty with respect to the archive paper.

---

> > ### Author Response · Authors · 2021-11-24
> > **Program Chair Response on Novelty**
> >
> > Dear Reviewer DomP,
> >
> > Our understanding is that your main criticism was about the novelty of our method given the existence of the SceneTransformer paper on arXiv.
> > Have the comments from the Program Chairs altered your assessment of the novelty of our method, and would you be willing to increase your score on those grounds?
> >
> > Thanks for your attention to this matter,
> > The Authors

---

> > > ### Comment · Reviewer_DomP · 2021-11-27
> > > **Thank you!**
> > >
> > > Sorry I wasn't sure where I found my concurrent work definition for ICLR'22. As you mentioned, my main concern was the novelty against SceneTransformer. If SceneTransformer is considered concurrent work, I have no issues to accept this paper. I'll fix my ratings.
> > >
> > > PS: Sorry for the delayed response.

---

> ### Author Response · Authors · 2021-11-17
> **Response to Reviewer DomP**
>
> We thank the reviewer for their valuable feedback on our manuscript. We are happy that the reviewer appreciates the extensive experiments and results we provided. We hope that the additional results presented in our general response to all reviewers are also appreciated.
>
> *Scene Transformer is very similar. Although SceneTransformer does not have an object query that is the latent variable Z, it was still able to learn a diverse set of future predictions. It's unclear if the explicit formulation of latent variable makes a difference.*
>
> In the appendix of the SceneTransformer paper, they explain that they concatenate a one-hot vector to the hidden representation of the encoder's output tensor. This allows the decoder to generate multiple diverse futures, one for each one-hot vector. During training, SceneTransformer backpropagates to the generated future that is closest to the ground-truth (i.e., the most likely mode receives a gradient signal). Although this is mathematically different, it is practically equivalent to using discrete latent variables, where each one-hot vector corresponds to a latent variable. The only difference is that our approach can choose to backpropagate softly to multiple modes that are likely instead of only the single most likely one.
>
> This use of one-hot vectors was the first approach we employed to represent the discrete latent variables of Autobot. However, we moved away from this approach due to computational efficiency reasons. This is where we differ from SceneTransformer. One of the main reasons that Autobot is so fast is the use of the decoder seed parameters, which have a shape of $(d_k, T, c)$ (where $d_k$ is the model hidden size, $T$ is the prediction horizon, and $c$ is the number of modes being modelled). These parameters serve a dual purpose. The first is that they account for the diversity in future prediction, where each matrix $(d_k, T)$ corresponds to one setting of the discrete latent variable. Secondly, they contribute to the speed of Autobot by allowing it to perform inference on the entire scene with a single forward pass through the decoder. Prior methods usually resort to using auto-regressive generation in the decoder in order to ensure socially-consistent futures. Our use of seed parameters in the decoder offers a more computationally efficient approach to generating scene consistent futures. Even though our performance is similar on Argoverse, Autobot is significantly faster to train as presented in Section 4.2, and contains significantly less parameters than the SceneTransformer (2M parameters (ours) versus 15M parameters (SceneTransformer)).

---

### Official Review · Reviewer_DDxv · 2021-11-04

**Correctness:** 3
**Technical Novelty And Significance:** 3
**Empirical Novelty And Significance:** 3
**Recommendation:** 8
**Confidence:** 5

**Main Review:**

Paper Strengths

1. The proposed method using the transformer to model both social and temporal information in both the encoder and decoder makes sense. Also, the ability to generate scene-consistent predictions are verified with both qualitative and quantitative results on TrajNet

2. The computation requirement of the proposed method is very appealing compared to other transformer-based motion prediction methods, as explained in section 4.2. With experience in running other similar methods, training usually takes 2-3 days on top-performing GPUs, whereas the proposed method only requires training on 1080ti for 10 hours.

3. The experiments are extensive, where the proposed method has been evaluated on three motion prediction datasets and one stroke completion dataset

Paper Weaknesses

1. Why is the proposed method very fast compared to others? Although a summary of computing requirements is provided, it would be interesting to perform a deeper analysis as to why the proposed transformer-based method is faster than other transformer-based methods, smaller feature dimensions? The fewer number of blocks? This could be helpful to the community in the future to develop efficient and high-performance transformer-based prediction methods

2. The ablation experiment can be expanded. Currently, the only ablation experiment is done in Table 3 on TrajNet with two variations of removing social modeling. However, the TrajNet challenge is a bit outdated and data annotation is imprecise. Would be more convincing to perform the ablation on the nuScenes or Argoverse dataset. Also, there are other components of the proposed method that are not ablated. For example, how is the proposed transformer-based approach compared to RNN-based or Graph-based methods? How is the map affecting the performance? How are the seed parameters useful? How many time/social blocks are needed?

3. The novelty is a bit weak considering other works have similar transformer-based motion prediction methods, especially [A1] as mentioned in section 5. Even ignoring this “concurrent” work, other works including [A2, A3] also use stacked transformers to do both social and temporal modeling. It is incorrect to me that the paper claims [A2, A3] did not encode the time and social dimensions using transformers. It would be great if the differences with these works can be explained in a more detailed and precise way

4. The experiments on TrajNet and Omniglot are not very convincing, where no quantitative comparison is properly made with prior work. For example, on the TrajNet challenge, there are plenty of prior motion prediction methods that have been evaluated on this dataset, so it should be easy to have a table for comparison. For stroke completion, I am not very familiar with this dataset, but only comparing with a simple LSTM baseline is not convincing. Including experiments on more datasets is great, but would be good to show convincing experiments.

References

[A1] AgentFormer: Agent-Aware Transformers for Socio-Temporal Multi-Agent Forecasting. ICCV 2021

[A2] Spatio-Temporal Graph Transformer Networks for Pedestrian Trajectory Prediction. ECCV 2020

[A3] Multimodal Motion Prediction with Stacked Transformers. CVPR 2021


Post-rebuttal review

After carefully reading other reviews and the authors’ comments, I would like to increase my rating to a score of 8 -- accept good paper (or a score of 7 if there is such an option). The change to my score is mainly because I am satisfied with most of the authors’ comments to my concerns, which include: 1) the clarification on why the proposed method is much faster; 2) the expanded ablation experiments; 4) the clarification for experiments on Omnlglot. I strongly appreciate the authors' efforts on clarifying these questions and concerns!

As a result, the main comment left on my side is 3) the limited novelty compared to prior work. This does not mean that I find the authors’ reply unconvincing. In fact, it is quite clear and honest. But essentially, the main difference to prior work seems to be just the use of seed parameters that increases performance and also runtime speed, since other points are already proposed in prior work, including joint social-temporal modeling in encoders/decoders using transformers, and scene-consistent predictions. But I would like to emphasize that this relatively incremental novelty is fine to me, and personally, I would like to have a try on the seeing parameters proposed in this work. So it would be wonderful for the community if the code can be carefully documented and released (currently I did not find it in the supp or any anonymous URL). Finally, as a minor point, it would be great if the map or context image can be overlaid on the trajectories for the provided video results.

In short, if there is no other significant downside/concern pointed by other reviewers, I would like to stick to my rating above and recommend accepting this paper



**Summary Of The Paper:**

This paper proposes a transformer-based VAE model for motion prediction that can output multi-model and scene-consistent predictions. Specifically, the transformer is employed for modeling both social and temporal information (but separately, first temporal and then social and repeat). The proposed method achieves state-of-the-art performance on the nuScenes dataset and top performance on the Argoverse dataset

**Summary Of The Review:**

Novelty is slightly weak but empirical performance is strong and the method might be practically useful

---

> ### Author Response · Authors · 2021-11-17
> **Response to Reviewer DDxv (1/2)**
>
> We thank the reviewer for their valuable feedback on our manuscript. We are happy to hear that the reviewer found our model’s computational requirements appealing, and that the use of social and temporal information in both the encoder and decoder are useful for making scene-consistent predictions.
>
> *1) Why is the proposed method very fast compared to other Transformer motion prediction methods?*
>
> One of the main aspects that makes Autobot so fast is the use of the decoder seed parameters, which have a shape of $(d_k, T, c)$ (where $d_k$ is the model hidden size, $T$ is the prediction horizon, and $c$ is the number of modes being modelled). These parameters serve a dual purpose. The first is that they account for the diversity in future prediction, where each matrix $(d_k, T)$ corresponds to one setting of the discrete latent variable. Secondly, they contribute to the speed of Autobot by allowing it to perform inference on the entire scene with a single forward pass through the decoder without sequential sampling.
>
> Some prior approaches resort to autoregressive generation of the future scene, which allows them to perform some form of social attention after every timestep of future rollout. These methods have the advantage of being able to perform joint predictions, but at the cost of expensive inference. Other existing methods encode the past socially, and perform inference in a single forward pass using an MLP to produce the entire future sequence of the ego-agent. This approach is cheap for inference but hinders the model’s ability to produce socially consistent futures. Using the attention mechanism of transformers, Autobot combines the advantages of both approaches by deterministically decoding the future scene starting from each seed parameter matrix. We added a paragraph in Section 3.2 to provide the intuition on the decoder seed parameters.
>
> A further aspect that contributes to Autobot’s speed is that it uses a relatively small number of parameters. For both Nuscenes and Argoverse, the reported model contains approximately 2 million parameters which is small compared to other models (e.g., SceneTransformer reports having approximately 16 million parameters).
>
> *2) The ablation experiments should be expanded.*
>
> We note that  the results on the TrajNet dataset have the purpose of showcasing the utility of the social components of Autobot. However, as you point out, the annotations are imprecise on this dataset. We are replacing those results with the synthetic portion of the TrajNet++ dataset, with the results shown in the Table above. This partition of the dataset was precisely designed to challenge models on their ability to perform motion prediction in interactive scenarios (see appendix of [citation] for more details).
>
> Please see our comment to all reviewers with the ablations that you requested. Note that these ablation results are on the Nuscenes' validation set and all models were trained for an equal number of epochs. As one would expect, the no-map ablation performs poorly overall, and especially on the off-road rate.
>
> The seed parameters serve a dual purpose of corresponding to the discrete latent variable and making inference efficient. The ‘No latent variable’ version of our model is a model that has the same number of encoder and decoder layers but is auto-regressive, and does not have the hidden variable. That is, during decoding, we sample the next point from the predicted bivariate gaussian distribution and feed that position as input to the decoder. This process is repeated c times in order to generate multiple futures. We can see that the model’s performance without the latent variables is worse. In addition, we note that this model is much slower to train due to the autoregression (approximately 3 times as long).
>
> For the required number of encoder-decoder layers, we are providing ablation results on the TrajNet++ synthetic dataset, as presented in our general comment. We reiterate that we chose this dataset because of the interaction between agents, and it was more amenable to analysis of joint predictions compared to Argoverse and Nuscenes. The table shows how the number of social/temporal blocks in the encoder and/or decoder affect the performance across all four metrics. As we can see, Autobot’s joint predictions improve significantly as the number of decoder layers is increased. It also improves with additional encoder layers but not as significantly as increasing the number of decoder layers. This is expected since the decoder layers intuitively contribute more to social attention in the decoding process.

---

> > ### Author Response · Authors · 2021-11-17
> > **Response to Reviewer DDxv (2/2)**
> >
> > *3) Related concurrent work*
> >
> > We outline key differences and have updated our related work section accordingly.
> >
> > [A1] Agent former differs from our method mainly because they apply both social and temporal attention by flattening both features into a single dimension. To differentiate time and agent attention they apply a mask directly to the flattened features space. We, on the other hand, interlace consecutive social and time based attention which produces a jointly time and social encoded context. We also do not rely on the autoregressive strategy to predict multiple outputs. Instead, we rely on the transformation of a single seed parameter matrix which decodes all the future positions in a single pass. These differences had a significant impact on the performance as validated in our experiments. On the NuScenes benchmark, [A1] obtained an average displacement error for five predictions (ADE5) of 1.86 versus our reported error of 1.43. And, for ADE 10 (10 predictions), they reported an error of 1.45 versus our reported error of 1.05.
> >
> > [A2] Indeed, it also performs both social and temporal attention jointly and for all agents. We added a line clarifying that on the related work section . Some of the key differences are the usage of graph encoding and auto-regressive predictions. We use tensor inputs and a permutation invariant based encoding that has way faster inference time. Also, they perform experiments on the Trajnet benchmark only which does not directly consider the multi-modality existent on the problem. Thus, their method has little consideration on multimodal predictions and obtains multiple model outputs by concatenating a gaussian input on the transformer embedding. This can lead to different predictions but does not necessarily produce a diverse output.
> >
> > [A3] mmTransformer do indeed use social and temporal encoding using transformers but the predictions are made separately for each agent and no joint future for the scene is predicted. We contrast by doing joint prediction for all agents on the scene in a single pass. Further, mmTransformer predicts the target final region that a certain vehicle will reach, and this makes this approach more specialized for driving scenarios. The proposed method offers a general set based formulation. We also note that the region based target could potentially be incorporated into our method in the future.
> >
> >
> > *4) Omniglot and TrajNet experiments are not convincing.*
> >
> > We hope that our explanations above specific to TrajNet (and the switch to TrajNet++) provide a more coherent view of why we chose this dataset. Other arms length colleagues suggested that we should provide results on another domain to demonstrate the versatility of our general framework. For the Omniglot experiments, our goal was to show that Autobot can be used in settings other than motion prediction. We compare it with an LSTM augmented with social attention as this is similar to the popular social LSTM baseline. We note that it is not really possible to compare with other models without re-implementing them, especially since the results we present on this dataset are qualitative. We believe that our experiments on Nuscenes and Argoverse provide enough comparisons with other strong methods in the motion forecasting literature.

---

### Author Response · Authors · 2021-11-17
**General Response to All Reviewers**

Dear reviewers,

We thank the reviewers for their valuable feedback on our manuscript.

To begin, we would like to point out that prior work using TrajNet has not evaluated the quality of multi-agent predictions, i.e., using **scene-level** metrics like Scene-Level minADE etc. This is why we did not compare with prior work in our table. Nevertheless, we provide additional results below for the synthetic part of the TrajNet++ dataset [1], which was specifically designed to challenge multi-agent motion prediction methods on highly interactive scenarios with high probabilities of collisions between agents (see appendix of [1] for more details). In addition, we uploaded videos of six scenes from the validation set across the three model variants. We believe that this dataset is better suited for evaluating joint predictions and showcases the usefulness of Autobot’s social attention components. This table replaces the current Table 3 in the manuscript, and the associated text has been added as Section 4.3.
The original TrajNet results have been moved to appendix D.3.

| Model | Ego minADE(c=6) | Number of Collisions | Scene-Level minADE (c=6) | Scene-Level minFDE (c=6) |
| -------- | -------- | -------- | -------- | -------- |
| Linear Extrapolation | 0.439 | 2220 | 0.409 | 0.897 |
| AutoBot-AntiSocial | 0.196 | 1827 | 0.316 | 0.632 |
| AutoBot-Ego | 0.098 | 1144 | 0.214 | 0.431 |
| AutoBot | 0.095 | 139 | 0.128 | 0.234 |
*Numbers computed on 5451 validation scenes of TrajNet++ synthetic dataset.
**The ego metric is computed on the agent around which the scene has been normalized.

In addition to the above ablation study on social components of Autobot, as requested, below we provide additional ablations on various components of our model following your recommendations. The table below shows the model with ablated map information, ablated seed parameters and ablating the data augmentation of randomly mirroring all agent trajectories (and map) during training on the Nuscenes dataset. This table is now Table 6 in manuscript (appendix).

| Model | minADE (c=5) | minADE (c=10) | MissRate (c=5) | MissRate (c=10) |Off Road Rate |
| -------- | -------- | -------- | -------- | -------- | -------- |
| Autobot-Ego | 1.43 | 1.05 | 0.66 | 0.45 | 0.03 |
| AutoBot-Ego (No Map information) | 1.88 | 1.35 | 0.72 | 0.58 | 0.223 |
| AutoBot-Ego (No latent variable) | 1.77 | 1.34 | 0.76 | 0.64 | 0.07 |
| AutoBot-Ego (No data augmentation) | 1.47 | 1.08 | 0.66 | 0.47 | 0.03 |
* Please note that we provide more details about these ablation experiments in our detailed response to reviewer DDxv who had requested these.

Finally, we performed the following new ablation studies on the TrajNet++ synthetic dataset to show how the joint performance of Autobot is affected by the change in the number of encoder and decoder social/temporal layers. This table is now Table 8 in the manuscript (appendix).

| Model | Ego minADE(c=6) | Number of Collisions | Scene-Level minADE (c=6) | Scene-Level minFDE (c=6) |
| -------- | -------- | -------- | -------- | -------- |
| L_{enc=1,L_{dec}=1 | 0.108 | 273 | 0.152 | 0.281 |
| L_{enc}=2,L_{dec}=1 | 0.102 | 214 | 0.138 | 0.254 |
| L_{enc}=3,L_{dec}=1 | 0.099 | 227 | 0.135 | 0.245 |
| L_{enc}=1,L_{dec}=2 | 0.099 | 164 |  0.132| 0.248 |
| L_{enc}=1,L_{dec}=3 | 0.091 | 123 | 0.121 | 0.225 |
| L_{enc}=2,L_{dec}=2 | 0.095 | 139 | 0.128 | 0.234 |
| L_{enc}=3,L_{dec}=3 | 0.090 | 98 | 0.119 | 0.216 |
*Numbers computed on 5451 validation scenes of TrajNet++ synthetic dataset.
**The ego metric is computed on the agent around which the scene has been normalized.

We believe that these additional experiments help strengthen the paper and showcase the strengths of Autobot. We thank the reviewers for their recommendation to perform these ablations and improved experiments. We are adding these additional results to the manuscript.

We thank the reviewers for their time. We will also provide individualized responses.

The Authors

[1] Kothari, P., Kreiss, S., & Alahi, A. (2020). Human Trajectory Forecasting in Crowds: A Deep Learning Perspective. ArXiv, abs/2007.03639.

---

### Decision · Program_Chairs · 2022-01-20

**Decision:**

Accept (Spotlight)

**Comment:**

This paper studies the problem of motion prediction for multiple agents in a scene using transformer-based VAE like architecture. The paper received mixed reviews initially which generally tended towards borderline acceptance. All reviews appreciated extensive experiments but had some clarifications and requests for ablations. The authors provided a strong rebuttal that addressed many of the reviewers' concerns. The paper was discussed and all the reviewers updated their reviews in the post-rebuttal phase. Reviewers unanimously agree that the paper should be accepted. AC agrees with the reviewers and suggests strong acceptance. The authors are urged to incorporate reviewers' comments in the camera-ready.